

# Phytoplankton adaptation to steady or changing environments affects marine ecosystem functioning

Isabell Hochfeld[1], Jana Hinners[2]

[1]Institute of Marine Ecosystem and Fishery Science, Universität Hamburg, Palmaille 9, 22767 Hamburg, Germany
ORCID: 0000-0002-5705-0549

[2]Helmholtz-Zentrum Hereon, Max-Planck-Straße 1, 21502 Geesthacht, Germany
ORCID: 0000-0002-5145-2539

*Correspondence to:* Isabell Hochfeld (isabell.hochfeld@uni-hamburg.de)

**Abstract.** Global warming poses a major threat to marine ecosystems, which fulfill important functions for humans and the climate. Ecosystem models are therefore increasingly used to estimate future changes in the functioning of marine ecosystems. However, projections differ notably between models. We propose that a major uncertainty factor in current models is that they ignore the high adaptive potential of phytoplankton, key players in marine ecosystems. Here, we use a 0-dimensional evolutionary ecosystem model to study how phytoplankton adaptation can affect estimations of future ecosystem-level changes. We found that phytoplankton adaptation can notably change simulated ecosystem dynamics, with the effect depending on environmental conditions. In a steady environment, adaptation allows for a more efficient use of resources, which enhances primary production and related ecosystem functions. In a warming environment, on the contrary, adaptation mitigates dominance changes among functionally different taxa and consequently leads to weaker changes in related ecosystem functions. Our results demonstrate that by neglecting phytoplankton adaptation, models may systematically overestimate future changes in the functioning of marine ecosystems. Future work can build on our results and include evolutionary processes into more complex model environments.

## 1 Introduction

Global warming leads to a rapid reorganization of marine ecosystems, which poses a major threat to their functioning (Pecl et al., 2017). Since changes in the functioning of marine ecosystems directly impact humans and even feed back on the climate, understanding them is crucial (Pecl et al., 2017; Prentice et al., 2015). Ecosystem models have proven a valuable tool



in this regard, but projections differ notably between models (Laufkötter et al., 2015, 2016). Current models largely ignore the high adaptive potential of phytoplankton (Laufkötter et al., 2015, 2016; Munkes et al., 2021), which are key players in marine ecosystems (Litchman et al., 2015). Here, we fill this gap by using an evolutionary ecosystem model to study the effect of phytoplankton adaptation to global warming on projected changes in ecosystem functioning. We apply the model to the Baltic Sea, which is impacted by above-average levels of multiple stressors (Reusch et al., 2018).

Phytoplankton contribute about half of global photosynthesis (Field et al., 1998), form the base of the marine food web (Fenchel, 1988), drive biogeochemical cycles (Hutchins and Fu, 2017), and even feed back on ocean physics (Hense, 2007; Sathyendranath et al., 1991). Furthermore, phytoplankton-related ecosystem functioning feeds back on the climate, e.g., through changes in the export of atmospheric carbon into deeper water layers (biological carbon pump) (Basu and Mackey, 2018) or the planktonic production of dimethyl sulfide, which seeds cloud formation (Wingenter et al., 2007). However, due to global warming, the role of phytoplankton in marine ecosystems is changing.

Phytoplankton respond to global warming through changes in phenology, which are expressed, for example, in an earlier and prolonged blooming season in the Baltic Sea (Wasmund et al., 2019). The resulting mismatches with higher trophic levels like zooplankton and fish alter food web structures and may eventually lead to ecosystem-level changes (Asch et al., 2019; Edwards and Richardson, 2004; Winder and Schindler, 2004a). In addition, the poleward migration of phytoplankton causes changes in species composition and abundance (Poloczanska et al., 2013), which may additionally affect zooplankton and fish stocks (Fossheim et al., 2015). Indeed, fisheries are already impacted by warming-related changes (Peterson et al., 2017). Finally, warming and eutrophication promote harmful algal blooms, which pose a threat to animal and human health (Gobler et al., 2017; Paerl et al., 2015; Glibert et al., 2014). To conclude, ongoing global warming will lead to changes in phytoplankton and consequently, to changes in the functioning of marine ecosystems. Since these changes are expected to have a direct impact on human well-being and the climate (Pecl et al., 2017; Prentice et al., 2015), predicting them is of great importance.

Ecosystem models offer the possibility to assess future changes in ecosystem functioning. For example, ecosystem models can be integrated into global ocean circulation models to simulate future changes in net primary production on global scale, but models do not even agree on the direction of change (Laufkötter et al., 2015). Similarly, regional models for



the Baltic Sea cannot agree on the future development of cyanobacteria blooms regarding timing, concentrations, and nitrogen fixation (Hense et al., 2013; Meier et al., 2011; Neumann, 2010). These uncertainties can notably affect estimations of future ocean deoxygenation (Long et al., 2021), nutrient load (Reusch et al., 2018; Wasmund et al., 2001), and harmful algal bloom
dynamics (Hallegraeff, 2010; Paerl et al., 2015). To conclude, the validity of current model projections remains questionable. Since model projections form the base of political decision making (Intergovernmental Panel on Climate Change (IPCC), 2022; Meier et al., 2014), there is an urgent need to improve their informative value. A first step could be to identify the key processes that affect ecosystem functioning. One key process that is lacking in all models above
and similar models (Daewel and Schrum, 2013; Dzierzbicka-Głowacka et al., 2013; Savchuk, 2002) is the evolutionary adaptation of phytoplankton.

Owing to their large population sizes and short generation times, phytoplankton possess a high potential to adapt to environmental changes. Evolution experiments, observations, and resurrection experiments demonstrated that phytoplankton adaptation can be relevant on
perennial or even shorter time scales (Jin and Agustí, 2018; Irwin et al., 2015; Hattich et al., 2024). Due to the crucial role of phytoplankton in marine ecosystems, considering phytoplankton adaptation in models may notably change projected changes in ecosystem functioning (Ward et al., 2019).

Some ecosystem models already consider the evolutionary adaptation of phytoplankton.
So far, evolutionary ecosystem models have generally been used to study the spatial distribution and/or temporal evolution of different functional traits (Le Gland et al., 2021; Beckmann et al., 2019; Sauterey et al., 2017). Only a few evolutionary ecosystem models have already addressed questions related to ecosystem functioning. For example, Smith et al. (2016) identified a trade-off between phytoplankton size-diversity and productivity depending on the frequency of
environmental disturbance. Sauterey & Ward (2022) investigated drivers of phytoplankton C:N stoichiometry, which affects the efficiency of the biological carbon pump. Finally, Cherabier & Ferrière (2022) studied the effect of bacterial adaptation to global warming on the microbial loop and the resulting impact on primary production.

So far, however, no model has explicitly addressed the question of how phytoplankton
adaptation to global warming could affect the functioning of a marine ecosystem. A first step might be to estimate the effect of adaptation on warming-related changes in phytoplankton community composition. Different phytoplankton functional groups fulfill different functions in the ecosystem, for example, by contributing differently to the biological carbon pump



(sinking speed), the nitrogen cycle (nitrogen fixation), and the energy transfer to higher trophic
levels (food quality, susceptibility to predation) (Litchman et al., 2015). To our knowledge,
there is only one model to date that considers competition between multiple phytoplankton
functional groups and their adaptation to global warming simultaneously (Hochfeld and
Hinners, 2024). Using this model, Hochfeld & Hinners (2024) demonstrated that adaptation can
significantly reduce simulated phytoplankton responses to global warming in terms of changes
in bloom timing and relative taxa abundance. However, it has not been studied yet how
adaptation-related changes in phytoplankton responses may affect ecosystem functioning.

Here, we use a slightly modified version of the Hochfeld & Hinners (2024) model to
estimate for the first time how phytoplankton adaptation may affect warming-related changes
in different ecosystem functions, including primary production, secondary production, carbon
export, nitrogen fixation, and resource use efficiency (RUE). We apply the model to the Baltic
Sea, which is already impacted by above-average levels of warming, nutrient load, and
deoxygenation (Reusch et al., 2018). Due to the 0-dimensional setup of the model, we do not
evaluate absolute changes in ecosystem functions. Instead, we focus on how phytoplankton
adaptation may change the future contribution of primary production to these ecosystem
functions. Our study is a first step to improve model projections of future ecosystem-level
changes that future work can build upon.

## 2 Materials and Methods

### 2.1 Model description

To study how phytoplankton adaptation to global warming may affect simulated future changes
in ecosystem functioning, we have slightly extended the model from Hochfeld & Hinners
(2024). A more detailed description of the model is available in Hochfeld & Hinners (2024) and
the associated supplementary material. The model simulates the dynamics of phytoplankton,
zooplankton, dissolved inorganic nitrogen, and dead organic matter (detritus) in a 0-
dimensional framework (Fig. 1). Since we focus on phytoplankton and their functions in the
marine ecosystem, we resolve three different phytoplankton functional groups. Like Hochfeld
& Hinners (2024), we chose three of the most common functional groups in the Baltic Sea,
dinoflagellates, diatoms, and diazotrophic cyanobacteria, and represent each group by a
common taxon or by a complex of common taxa. For dinoflagellates and diatoms, we simulate
two cold-water species of the genera *Apocalathium* and *Thalassiosira*, respectively. For



cyanobacteria, we simulate a complex that represents the dominant nitrogen-fixing genera in the Baltic Sea, *Nodularia*, *Aphanizomenon*, and *Anabaena* (Karlsson et al., 2005; Stal et al., 2003). Like other modeling studies (Hense and Beckmann, 2006; Hinners et al., 2015; Lee et al., 2018), we assume cyanobacteria to be non-grazeable due to toxicity, while dinoflagellates and diatoms are equally grazed by zooplankton.

To ensure an accurate representation of phytoplankton phenology under warming conditions, the model explicitly resolves phytoplankton life cycle dynamics. For all functional groups, the model differentiates between a resting stage and a vegetative growing stage, with growth being limited by light, temperature, and dissolved inorganic nitrogen. The cyanobacteria life cycle additionally includes a diazotrophic growing stage, which can fix atmospheric nitrogen ($N_2$) and is therefore not limited by dissolved inorganic nitrogen. Hence, dissolved inorganic nitrogen is taken up by all phytoplankton growing stages except for the diazotrophic growing stage of cyanobacteria. The nitrogen content of all dead phytoplankton and zooplankton cells fills the detritus pool, which is remineralized back into bioavailable nitrogen at a constant rate. Due to sinking of detritus and stochastic burial of phytoplankton resting cells, nitrogen is lost from the system. Nitrogen can be replenished through the resuspension of previously buried resting cells and cyanobacterial nitrogen fixation.

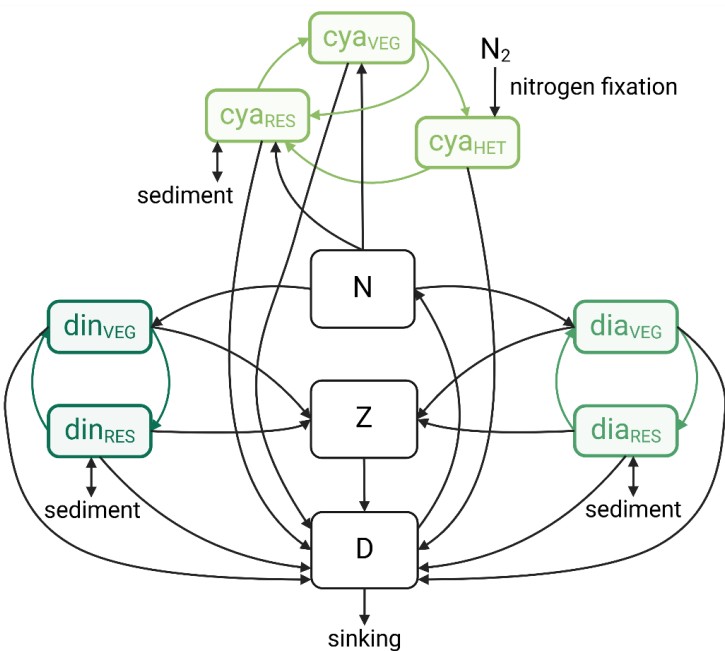



**Figure 1:** Components of the ecosystem model including compartments for dissolved inorganic nitrogen (N), detritus (D), and zooplankton (Z), along with agent-based life cycles of dinoflagellates (din), diatoms (dia), and cyanobacteria (cya). Each life cycle is represented by a resting stage (RES) and a vegetative growing stage (vegetative cells, VEG). For cyanobacteria, the model simulates a second, nitrogen-fixing growing stage (vegetative cells with heterocysts, HET). The figure additionally shows the nitrogen fluxes between the different ecosystem components, and the sinks and sources of nitrogen (sinking of detritus, burial of phytoplankton resting cells and resuspension of phytoplankton resting cells, cyanobacterial nitrogen fixation). The figure was adapted from Hochfeld & Hinners (2024) and created with BioRender.com.

The model does not only consider competition for nitrogen between different phytoplankton taxa, but also changes in two temperature-dependent functional traits. The first flexible trait, the optimum temperature for growth, adapts through random mutations. Cell size, on the contrary, responds plastically to temperature, with the cell size decreasing linearly with increasing temperature (Atkinson et al., 2003). For further details on the implementation of mutations and plasticity, see Hochfeld & Hinners (2024). The model additionally considers that changes in cell size affect metabolic cell properties (Litchman et al., 2007; Marañón et al., 2013; Ward et al., 2017), which in turn determine the nitrogen-limited growth rate (Grover, 1991). Since trait changes such as those described above affect individual cells, the model uses an agent-based approach after Beckmann et al. (2019) to simulate the dynamics of agents (super-individuals) with their individual phenotypic trait values. Zooplankton, dissolved inorganic nitrogen, and detritus, on the contrary, are represented by compartments, i.e., collections of cells or molecules described by their averaged properties and their concentration.

## 2.2 Ecosystem functions

Our extended version of the Hochfeld & Hinners (2024) model allows us to analyze different ecosystem functions, including carbon export, cyanobacterial nitrogen fixation, and resource use efficiency (RUE).

We calculate carbon export from the carbon content of buried phytoplankton resting cells and the carbon that is exported through sinking of detritus. Detritus contains the dead phytoplankton and zooplankton cells, as well as the remains from unassimilated feeding. Following Ward et al. (2012), we divide detritus into dissolved inorganic matter (DOM) and particulate organic matter (POM), of which only POM is exported into deeper water layers. For



the taxonomic groups in our model, we divide detritus 50:50 between POM and DOM (Ward et al., 2012). Since the model calculates in nitrogen units, we use the Redfield ratio to convert nitrogen into carbon (Redfield, 1934).

To determine the amount of fixed atmospheric nitrogen, we assume that all the fixed nitrogen is converted into biomass. Thus, we define nitrogen fixation as the biomass built up by the diazotrophic cyanobacteria life cycle stage during each time step.

     Following Ptacnik et al. (2008), we calculate resource use efficiency (RUE) as the ratio of phytoplankton biomass and dissolved inorganic nitrogen. Since the cyanobacteria in our model can fix atmospheric nitrogen, we use simulations without cyanobacteria to derive RUE.
Hence, we only consider the RUE of dinoflagellates and diatoms. Both dinoflagellates and diatoms are grazed by zooplankton; to avoid grazing-related biases in RUE, we additionally exclude zooplankton from RUE simulations.

### 200    2.3 Model scenarios

To understand how the adaptation of phytoplankton to different environments affects model estimations of related ecosystem functions, we implement four different model scenarios based on Hochfeld & Hinners (2024) (Table 1). We perform seven simulations for each scenario and average the output to ensure robust results. Each simulation is run over 100 years.

The first two model scenarios C (control) and CA (control and adaptation) represent control scenarios, which we force with a steady seasonal temperature and irradiance forcing for present-day conditions in the Gulf of Finland. We use the same forcing as Hochfeld & Hinners (2024). The two control scenarios C and CA serve as spin up for two global warming scenarios W (warming) and WA (warming and adaptation). We simulate global warming by adding a
steady temperature increase of 0.3 °C per decade to the seasonal temperature forcing, which corresponds to the IPCC scenario SSP3-7.0 (Allan et al., 2021). While adaptation in the optimum temperature is disabled in C and W, it is enabled in CA and WA. In this way, we can study how the (in)ability of phytoplankton to adapt to their environment may affect ecosystem functioning.

In the four model scenarios presented above, the resuspension of phytoplankton resting cells from the sediment is disabled. Hochfeld & Hinners (2024) found that resuspension tends to slow down adaptation to global warming and can hence weaken adaptation-related effects. For completeness, we performed additional control and warming simulations in which we





enabled resuspension (CAR: control, adaptation, and resuspension, WAR: warming, adaptation,
and resuspension) and observed a similar effect. Thus, we do not explicitly analyze and discuss
these simulations here; an example is shown in Fig. B1.

**Table 1:** Overview of the four model scenarios that we evaluate in this article. For each scenario, we
run seven different simulations over 100 years and average the output. *Control* represents a present-day
seasonal temperature forcing for the Gulf of Finland. *Warming* adds a constant temperature increase of
0.3 °C per decade to the control forcing (IPCC scenario SSP3-7.0, Allan et al., 2021).

|           | No adaptation | Adaptation |
|-----------|---------------|------------|
| Control   | C             | CA         |
| Warming   | W             | WA         |

## 3 Results

### 3.1 Seasonal phytoplankton dynamics

The seasonal phytoplankton dynamics and the reasons for differences between scenarios are
described in detail in Hochfeld & Hinners (2024). In summary, the two control scenarios C and
CA produce a realistic seasonal cycle for the focal phytoplankton taxa, including a spring bloom
of dinoflagellates and diatoms, a summer bloom of cyanobacteria, and a second but weaker
bloom of diatoms in autumn (Fig. 2). In CA, where phytoplankton can adapt, competition for
nitrogen drives adaptation to individual temperature niches, which reduces competition
pressure. Due to reduced competition with diatoms, cyanobacteria can initiate a stronger
summer bloom, which increases the amount of nitrogen in the system through nitrogen fixation.
The higher concentration of nitrogen, in turn, allows for stronger blooms of dinoflagellates and
diatoms.

The two warming scenarios W and WA were found to reproduce trends that have been
observed in the Baltic Sea over the past decades (Hochfeld and Hinners, 2024), including an
earlier and prolonged phytoplankton blooming season (Wasmund et al., 2019) as well as an
increase in cyanobacterial summer biomass (Suikkanen et al., 2007). The warming-related
changes in bloom timing and cyanobacteria biomass were shown to be weaker in the presence
of adaptation by up to ~9 d and 56 %, respectively (Hochfeld and Hinners, 2024). Adaptation
to the increasing temperatures in WA enhances the competitivity of non-pre-adapted taxa. Thus,





non-pre-adapted diatoms can compete more strongly with pre-adapted cyanobacteria, which leads to a weaker cyanobacterial summer bloom (Hochfeld and Hinners, 2024).

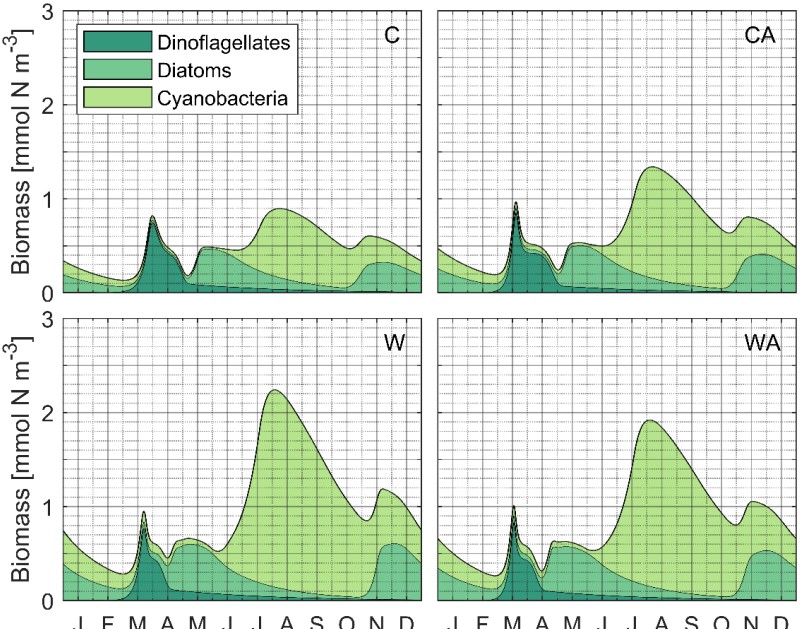

**Figure 2:** Accumulated phytoplankton biomass during the last simulation year of the four different model scenarios (C: control, CA: control and adaptation, W: warming, WA: warming and adaptation). For each scenario, the output of seven different simulations was averaged. The colors indicate the share of dinoflagellates, diatoms, and cyanobacteria in the total phytoplankton biomass.


### 3.2 Seasonal zooplankton dynamics

For all model scenarios, zooplankton biomass peaks during phytoplankton spring bloom following the peak in phytoplankton biomass; remember that we simulate cyanobacteria as single summer-blooming phytoplankton taxon, which we assume to be non-grazeable due to
toxicity. Despite these simplifications in the model, the simulated seasonal pattern is indeed reasonable for some of the common zooplankton taxa in the Baltic Sea (Feike et al., 2007; Dutz et al., 2010). Although all four model scenarios agree on a general seasonal pattern, both bloom timing and amplitude differ notably between them (Fig. 3 and Table 2), with the differences being statistically significant according to a *t*-test (Table A1).



In the control scenario with phytoplankton adaptation, CA, we observe an earlier and stronger zooplankton bloom than in the control scenario without phytoplankton adaptation, C (Fig. 3). In CA, zooplankton peak ~16 d earlier than in C with a ~52 % higher peak amplitude (Table 2). These findings resemble the dynamics of phytoplankton under control conditions, which develop an earlier and stronger spring bloom if they can adapt (Fig. 2).

Likewise, zooplankton show similar responses as phytoplankton to global warming, including a shift in bloom timing towards winter and an increase in peak amplitude, with the responses being weaker when phytoplankton adaptation is enabled (Fig. 3). While the zooplankton spring bloom peaks ~17 d and ~5 d earlier in W and WA, bloom amplitude increases by ~92 % and ~21 %, respectively (Table 2). In conclusion, zooplankton strongly

resemble the dynamics of phytoplankton in all four model scenarios.

        Irrespective of these similarities between phytoplankton and zooplankton, however, the time lag between their bloom peaks differs notably between the four model scenarios. Under control conditions, we observe a time lag of ~13 d and ~8 d in C and CA, respectively (Table 2). The two warming scenarios W and WA, on the contrary, produce a comparable and notably

shorter time lag of only ~4 d. Thus, in our simulations, warming seems to reduce the time lag between phytoplankton and zooplankton blooms. In addition, we find that the time lag correlates negatively with the peak amplitudes of both phytoplankton and zooplankton, meaning that the higher the amplitude, the shorter the time lag (Fig. B2). While both correlations are significant, the time lag correlates notably stronger with zooplankton peak

amplitude than with phytoplankton peak amplitude ($r(26) = -0.99$, $p = 2.05 \times 10^{-21}$ for zooplankton and $r(26) = -0.81$, $p = 2.41 \times 10^{-7}$ for phytoplankton).

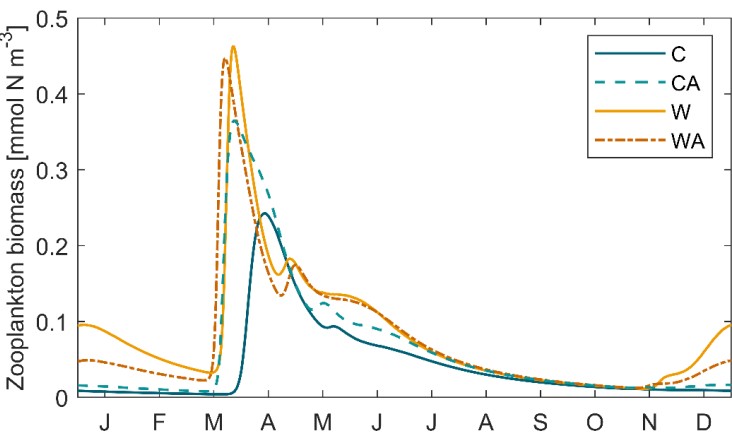





**Figure 3:** Zooplankton biomass during the last simulation year of the four different model scenarios (C: control, CA: control and adaptation, W: warming, WA: warming and adaptation). For each scenario, we
averaged the output of seven different simulations.

**Table 2:** Average zooplankton timing, peak abundance, and time lag to the phytoplankton peak in spring for the two control scenarios C (control) and CA (control and adaptation), along with the associated standard deviations. Also shown are the corresponding average warming-related changes in W
(warming), and WA (warming and adaptation), including propagated errors. For each scenario, we calculated average values from the last simulation year of seven different simulations. Please note that warming-related changes in zooplankton peak abundance are not presented as absolute values but as relative changes. A series of *t*-tests revealed that the differences between all four scenarios are statistically significant at the 0.05 level; see Table A1 for details.

|  | C | CA | W | WA |
|---|---|---|---|---|
| Timing [d] | 102.4 ± 2.0 | 86.5 ± 2.4 | -16.9 ± 2.3 | -5.5 ± 2.6 |
| Abundance [$\mu$mol N m$^{-3}$] | 244.8 ± 17.2 | 373.0 ± 46.0 | +92.4 % ± 7.4 % | +21.2 % ± 13.0 % |
| Time lag [d] | 12.9 ± 2.4 | 8.0 ± 2.5 | -8.7 ± 2.8 | -3.5 ± 2.8 |


## 3.3 Annual balances

The annual balances of different ecosystem functions are shown in Fig. 4 for the last simulation year of all model scenarios. Figure 4 reveals that phytoplankton produce ~10 times more biomass than zooplankton per year and hence dominate biomass production in our simulations.
Primary production, in turn, is dominated by cyanobacteria, while dinoflagellates account for the smallest amount of annual primary production.





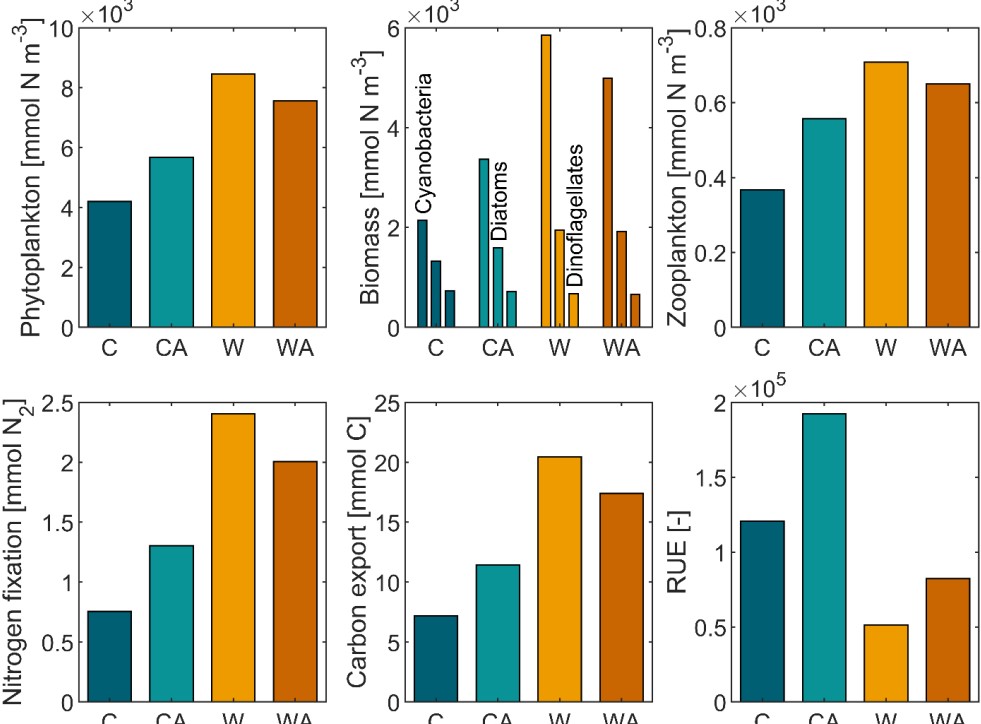

**Figure 4:** Annual balances of different ecosystem functions for the last simulation year of the four model scenarios (C: control, CA: control and adaptation, W: warming, WA: warming and adaptation). For each scenario, annual balances were averaged from seven different simulations.

For cyanobacteria, annual biomass increases under global warming, with the increase being by ~56 % weaker if thermal adaptation is enabled (Table 3). Under control conditions, on the contrary, cyanobacteria biomass is by ~52 % higher with thermal adaptation. While diatoms follow a similar trend, however with smaller differences between the scenarios, dinoflagellates show a contrasting development. Dinoflagellate annual biomass decreases slightly under global warming and is comparable between C and CA as well as W and WA, respectively. Thus, thermal adaptation does not seem to have a notable effect on the biomass production of dinoflagellates. Despite the contrasting development of dinoflagellates, total phytoplankton biomass follows the same trend as cyanobacteria and diatoms. This finding is underlined by strong positive correlations between total phytoplankton biomass, cyanobacteria, and diatoms, while dinoflagellates correlate negatively with all three (Fig. 5). In all four model scenarios,





total phytoplankton biomass correlates strongest with cyanobacteria ($0.98 \leq r \leq 1$) and weakest with dinoflagellates ($-0.73 \leq r \leq -0.07$).

Zooplankton annual biomass also correlates positively with the annual biomasses of diatoms, cyanobacteria, and total phytoplankton. Under control conditions, correlation is strongest with diatoms ($r \geq 0.81$), while under global warming, zooplankton biomass correlates strongest with total phytoplankton biomass ($r \geq 0.60$). In addition, zooplankton biomass production is notably affected by phytoplankton adaptation, which is consistent with our findings from the previous section. Under control conditions, zooplankton produce by ~52 % more biomass if phytoplankton can adapt. Under global warming, zooplankton biomass increases, with the increase being by ~73 % weaker when phytoplankton adaptation is enabled.

The annual amount of fixed atmospheric nitrogen mirrors the annual biomass of cyanobacteria, which is confirmed by a strong positive correlation in all four model scenarios with $r \geq 0.99$. Under control conditions, cyanobacteria fix ~72 % more nitrogen when adaptation is enabled. Global warming leads to an increase in nitrogen fixation, and hence the nitrogen input into the system, by ~218 % in W and ~54 % in WA, respectively.

Carbon export correlates positively with both phytoplankton and zooplankton biomass, with the correlation being stronger with phytoplankton, which dominate biomass production ($r \geq 0.90$ vs. $r \geq 0.69$). Among phytoplankton, carbon export correlates strongest with cyanobacteria, which dominate primary production ($r \geq 0.85$). In addition, carbon export is notably affected by phytoplankton adaptation. Under present-day conditions, carbon export is by ~59 % higher in CA than in C. Global warming leads to an increase in carbon export by ~184 % in W and ~52 % in WA, respectively.

Finally, resource use efficiency (RUE) decreases under global warming in our simulations, with the decrease being similar with and without phytoplankton adaptation (~57 % and ~58 %, respectively). Independent of the climate scenario, RUE is always higher if phytoplankton can adapt. Phytoplankton adaptation leads to an increase in RUE by ~59 % and ~61 % under control and warming conditions, respectively.

In conclusion, all ecosystem functions that we investigate in this study, except for dinoflagellates, show similar developments in the four model scenarios. This is underlined by strong positive correlations, which are significant at the 0.05 level (Fig. B3). Dinoflagellates, on the contrary, correlate (mostly) negatively with all other ecosystem functions; correlations with dinoflagellates are only partly significant, though. Independent of their direction, all





correlations notably change their strength between the four model scenarios. Under control conditions, all correlations are stronger if phytoplankton adaptation is enabled. This pattern reverses under global warming, where correlations are weaker with adaptation. This weakening is particularly strong for zooplankton, for which the negative correlation with dinoflagellates turns slightly positive in WA.

**Table 3:** Average annual balances for the two control scenarios C (control) and CA (control and adaptation), along with the associated standard deviations. Additionally shown are the corresponding average warming-related changes in W (warming), and WA (warming and adaptation), including propagated errors. For each scenario, we calculated average values from the last simulation year of seven different simulations. Please note that the warming-related changes in W and WA are not presented as absolute values but as relative changes. A series of *t*-tests demonstrated that the differences between all four model scenarios are statistically significant at the 0.05 level with only one exception (dinoflagellates in W and WA). See Table A2 for details.

|  | C | CA | W [%] | WA [%] |
|---|---|---|---|---|
| Dinoflagellates [mmol N m$^{-3}$] | 732.7 ± 9.2 | 718.1 ± 15.1 | -8.6 ± 2.4 | -8.8 ± 3.0 |
| Diatoms [mmol N m$^{-3}$] | 1327.6 ± 44.0 | 1591.8 ± 78.5 | +46.3 ± 3.4 | +20.2 ± 5.1 |
| Cyanobacteria [mmol N m$^{-3}$] | 2140.8 ± 80.9 | 3367.2 ± 524.8 | +173.3 ± 3.9 | +48.2 ± 15.9 |
| Phytoplankton [mmol N m$^{-3}$] | 4201.1 ± 121.9 | 5677.0 ± 597.2 | +101.4 ± 3.0 | +33.2 ±10.8 |
| Zooplankton [mmol N m$^{-3}$] | 366.9 ± 19.7 | 556.7 ± 51.0 | +92.9 ± 5.5 | +16.8 ± 9.6 |
| N$_2$ fixation [µmol N$_2$ m$^{-3}$] | 755.6 ± 40.3 | 1303.3 ± 246.3 | +218.1 ± 5.4 | +53.8 ± 19.4 |
| Carbon export [µmol C m$^{-3}$] | 7194.9 ± 356.3 | 11429.0 ± 1868.9 | +184.1 ± 5.0 | +52.3 ± 16.7 |
| RUE [$10^3$] | 120.6 ± 3.7 | 192.3 ± 5.9 | -57.6 ± 4.1 | -57.2 ± 4.7 |



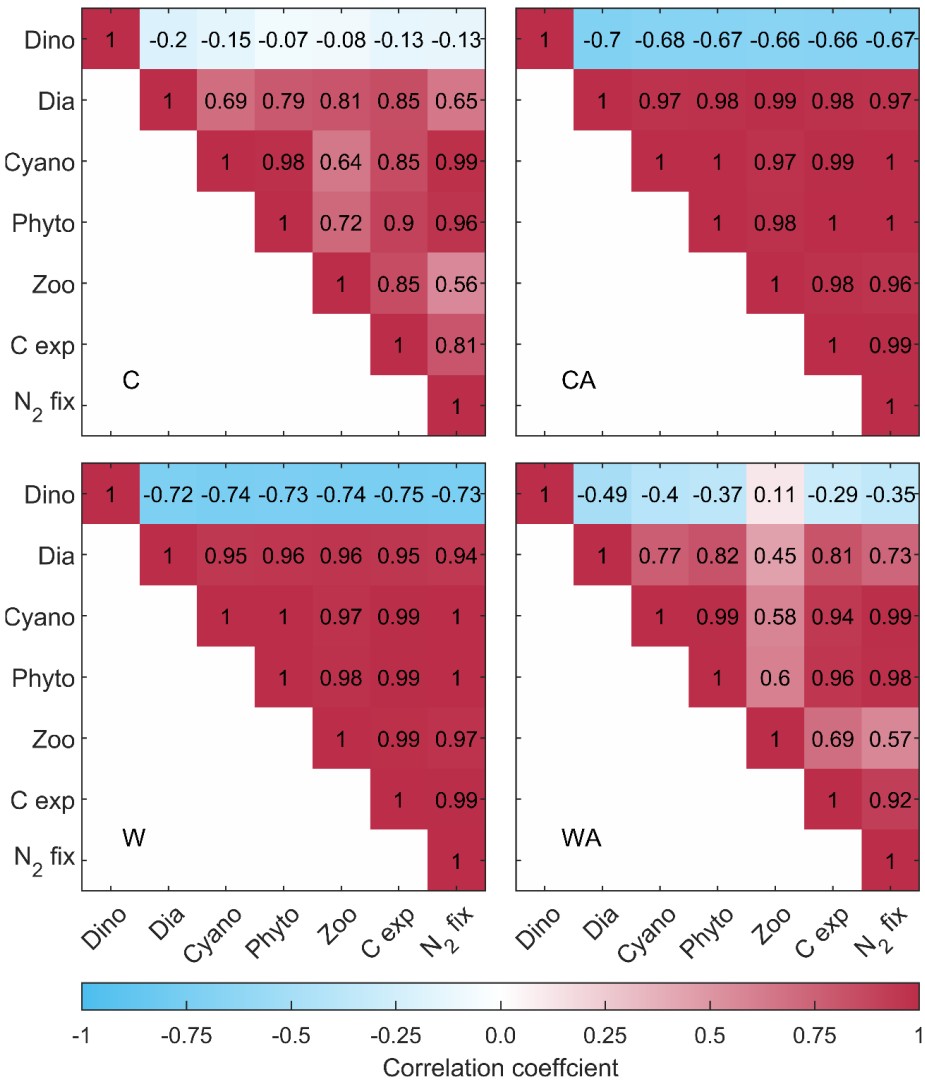

**Figure 5:** Correlation matrices showing the correlation coefficients between different ecosystem functions for the four different model scenarios (C: control, CA: control and adaptation, W: warming, WA: warming and adaptation). For C and CA, we calculated correlation coefficients using the annual balances from the last 95 years of seven different simulations. For W and WA, however, we only used the last 40 years to capture warming-related changes. All correlations shown here, except for those with dinoflagellates, are significant at the 0.05 level according to a *t*-test (see Fig. B3). Please note that resource use efficiency (RUE) is not included since we derived RUE from simulations without cyanobacteria and zooplankton.



## 4 Discussion

In this study, we used an evolutionary ecosystem model to analyze how ecosystem functioning may change in response to global warming, and how these changes may be affected by phytoplankton adaptation. We found that phytoplankton and zooplankton respond similarly to global warming, with the responses being weaker in the presence of phytoplankton adaptation. Likewise, warming-induced changes in associated ecosystem functions are generally less pronounced if phytoplankton adaptation is enabled in our simulations.

### 4.1 Phytoplankton

The model projects an increase in total phytoplankton biomass in response to global warming. This increase is predominantly driven by cyanobacteria, which are pre-adapted to high temperatures (Collins and Boylen, 1982; Lehtimäki et al., 1997; Nalewajko and Murphy, 2001). This finding agrees with observations, which have revealed a strong increase in cyanobacterial summer biomass in the Baltic Sea over the past decades (Suikkanen et al., 2007). A further increase in cyanobacteria in the future can have severe consequences for the ecosystem, for example, due to their toxicity for higher trophic levels (Repavich et al., 1990; Quesada et al., 2006; Chorus and Welker, 2021) and their ability to fix atmospheric nitrogen. We discuss potential impacts of increasing nitrogen fixation in Sect. 4.4. Future work can build on our results by including an explicit representation of cyanotoxin production and its effects on higher trophic levels.

While diatoms also increase under global warming, dinoflagellates show a slight warming-related decrease in annual biomass. This finding disagrees with observations, which report a shift from diatom to dinoflagellate dominance during spring bloom over the past decades in several areas of the Baltic Sea (Klais et al., 2011). These observations, however, are on functional group level, while we simulate one focal species per group. Resurrection experiments with our focal cold-water dinoflagellate of the genus *Apocalathium* revealed that encystment strongly depends on temperature, and that the temperature threshold for encystment remained constant over the past century of global warming at around 6 °C (Hinners et al., 2017). However, experiments by Kremp et al. (2009) showed that encystment strategies vary among Baltic cold-water dinoflagellates, with temperature not always being the main trigger mechanism. Thus, our model may not be appropriate for estimating future changes in overall spring bloom dynamics but only changes in our focal species.



Considering the dinoflagellate *Apocalathium* specifically, warming leads to an earlier onset of encystment and hence an earlier termination of the spring bloom. As a result, less cysts are produced and the inoculum decreases, weakening the spring bloom of *Apocalathium* over the years as global warming progresses. Consequently, our simulations suggest that warming induces negative feedback in the life cycle of *Apocalathium*. However, Hinners et al. (2017) found that *Apocalathium* has decreased its encystment rate over the past century of global warming , which prevents an abrupt bloom termination at temperatures around 6 °C. To test if a decrease in the encystment rate could weaken the negative feedback in our simulations, we performed additional simulations in which we artificially decreased *Apocalathium*'s encystment rate at the rate measured by Hinners et al. (2017). The simulations reveal that a corresponding decrease in the encystment rate leads to an even stronger decrease in the biomass of *Apocalathium* under global warming (Fig. B4). This suggests that the encystment rate of *Apocalathium* may respond differently to future climate change than to past climate change, or that we are missing another crucial factor. Further research is needed, for example in the form of evolution experiments. In addition, future work can build on our model and include an explicit representation of adaptation in the encystment rate of *Apocalathium*.

On the contrary to *Apocalathium*, our focal cold-water diatom of the genus *Thalassiosira* benefits indirectly from warming due to the increase in cyanobacterial nitrogen fixation. The more nitrogen is fixed in summer, the stronger is the bloom of *Thalassiosira* in autumn. A stronger autumn bloom adds more spores to the inoculum, and a larger inoculum allows for a stronger bloom of *Thalassiosira* in spring, which is further promoted by the weaker bloom of *Apocalathium*. The stronger spring bloom of *Thalassiosira* further increases the inoculum pool, which, in turn, further enhances the autumn bloom. Thus, on the contrary to *Apocalathium*, warming indirectly induces positive feedback in the life cycle of *Thalassiosira*, which is mainly driven by the response of cyanobacteria and, to a lesser extent, by that of *Apocalathium*.

To conclude, our results demonstrate that the responses of different phytoplankton taxa affect each other due to differences in their physiology and function. Thus, when simulating ecosystem-level responses to changing environments, it is crucial for models to include functionally different taxa with their individual physiologies (e.g., life cycle dynamics) to account for feedback and competition. As already demonstrated by Hochfeld & Hinners (2024), an adequate representation of competition also requires an explicit simulation of evolutionary adaptation.



### 4.2 Zooplankton

Our simulated zooplankton responses to global warming qualitatively agree with our simulated responses of phytoplankton; in both cases, responses are weaker if phytoplankton adaptation is
enabled. In our warming scenarios, both phytoplankton and zooplankton increase in abundance. A study by Richardson & Shoeman (2004) demonstrated that the abundance of herbivorous zooplankton significantly depends on their phytoplankton prey (bottom-up control), meaning that a warming-related increase in phytoplankton will most likely lead to an increase in zooplankton abundance.

In addition, our simulations show a warming-related shift in bloom timing towards winter for both phytoplankton and zooplankton, with the shift being stronger for zooplankton. Hence, our model does not produce a warming-related increase in the time lag between phytoplankton and zooplankton blooms as suggested by several studies (Edwards and Richardson, 2004; Winder and Schindler, 2004a, b; Adrian et al., 2006). Instead, the time lag
between phytoplankton and zooplankton tends to decrease in our warming scenarios compared to the corresponding control scenarios. This decrease in time lag is strongly connected to zooplankton peak amplitude, with higher peak amplitudes coinciding with shorter time lags. Higher zooplankton peak amplitudes indicate stronger grazing on phytoplankton, and hence stronger top-down control. This means that the time lag between phytoplankton and
zooplankton seems to decrease when top-down control increases.

The decreasing time lag in our simulations may result from our simplistic representation of zooplankton. We assume that zooplankton grazing depends exclusively on phytoplankton biomass and do not consider potential effects of irradiance and temperature. Moreover, we neglect both zooplankton life cycle dynamics and adaptation. However, observations show that
several zooplankton taxa peak earlier in the season in response to global warming (Richardson, 2008). Dam (2013) interprets the observed phenological shifts in zooplankton as a combination of ecological and evolutionary responses. For example, Dam (2013) argues that zooplankton do not only respond to changes in temperature itself but also to phenological changes in prey, which select for fast-growing zooplankton. Indeed, some phytoplankton and zooplankton taxa
show synchronous shifts in bloom timing, for example diatoms and *Daphnia* (Adrian et al., 2006). Some studies even suggest a warming-related decrease in the time lag between phytoplankton and zooplankton (Aberle et al., 2012; Almén and Tamelander, 2020). Consequently, the reduced time lag produced by our model indeed seems realistic for fast





growing zooplankton taxa like *Daphnia*, which are "selected" in our global warming
simulations by the earlier and stronger phytoplankton spring bloom. However, our model is not
suitable for simulating slow-growing zooplankton with longer and more complex life cycles
such as copepods or larvae of the mussel *Dreissena polymorpha* (Adrian et al., 2006).

In conclusion, our results suggest that warming-related responses of fast-growing
zooplankton may be closely related to responses of their phytoplankton prey. Thus,
phytoplankton adaptation may indeed reduce zooplankton responses to global warming, and
the effects of phytoplankton adaptation may even propagate further up the food chain. Future
work can build on our model and study how a more complex representation of zooplankton,
including both fast- and slow-growing taxa, and higher trophic levels may be affected by
phytoplankton adaptation.

### 4.3 Carbon export

Our simulations project a warming-related increase in carbon export in the future, which is
more than halved if phytoplankton adaptation is enabled. The projected changes in carbon
export correlate significantly with projected changes in biomass production, which are
dominated by a strong increase in cyanobacterial summer biomass. In the Baltic Sea,
cyanobacteria blooms have intensified over the last century of global warming (Finni et al.,
2001), especially during the last decades (Suikkanen et al., 2007). This development is reflected
by sediment records, which show a simultaneous increase in cyanobacteria pigments and carbon
content during the same period (Poutanen and Nikkilä, 2001). In the future, warming is
expected to further increase summer primary production with a positive feedback on carbon
export in several areas of the Baltic Sea (Tamelander et al., 2017).

Even if our model results are consistent with these findings, we need to keep in mind
that we use a 0-dimensional model setup, which cannot represent certain mechanisms that are
crucial for carbon export. For example, we cannot explicitly simulate physical processes in the
ocean like vertical mixing, including seasonal changes in stratification and mixed layer depth.
In addition, crucial processes like gravitational particle sinking and fragmentation are only
included implicitly in our model, while we neglect vertical migration of zooplankton and nekton
(Henson et al., 2022). Finally, in semi-enclosed ecosystems like the Baltic Sea, carbon export
is not predominantly fueled by phytoplankton primary production but also by benthic primary
production and riverine and terrestrial inputs (Goñi et al., 2000; Renaud et al., 2015; Tallberg



and Heiskanen, 1998). Since these key processes (and maybe others) are lacking in our model, we cannot interpret our results as projections of future carbon export. Instead, we interpret them as projections of the future contribution of primary production to carbon export. Our results reveal that the contribution of primary production to carbon export may increase in the Baltic

Sea in the future and that phytoplankton adaptation may notably weaken this increase.

### 4.4 Nitrogen fixation

Our model results suggest a strong warming-related increase in nitrogen fixation in the future, which is a direct result of the projected increase in cyanobacterial summer biomass. Today, the

Baltic Sea is already impacted by above-average levels of nutrient load (Reusch et al., 2018). For example, nitrogen-driven eutrophication turned the Baltic Sea into one of the most hypoxic ocean areas worldwide, with severe consequences for productivity, biodiversity, and biogeochemical cycling (Breitburg et al., 2018). In the future, global warming is expected to further increase the vulnerability of coastal systems to nutrient loading as harmful algal bloom

events become more likely and pose an increasing threat to animal and human health (Gobler et al., 2017; Paerl et al., 2015; Glibert et al., 2014).

Since the 1970s, nutrient management strategies have been applied to the Baltic Sea catchment area, resulting in a reduction of anthropogenic nitrogen load by ~25 % (Reusch et al., 2018). At the same time, however, nitrogen load by fixation increased notably (Gustafsson

et al., 2017). Model simulations demonstrated that the contribution of nitrogen fixation to the total nitrogen load to the Baltic Sea increased from almost 20 % in the 1980s to almost 35 % in the 2000s, so that the total nitrogen load decreased by only ~9 % (Gustafsson et al., 2017). For the future, our results suggest that the importance of cyanobacterial nitrogen fixation for the nitrogen budget of the Baltic Sea will most likely continue to increase and further mitigate the

success of nutrient management strategies. Therefore, nutrient management strategies urgently need to account for nitrogen load by fixation to be successful in the future. Since our projected increase in nitrogen fixation is more than halved if we consider phytoplankton adaptation, we strongly recommend that models used for assessment consider phytoplankton adaptation to realistically estimate future nitrogen load by fixation.





### 4.5 Resource use efficiency (RUE)

Since we had to exclude nitrogen-fixing cyanobacteria from RUE simulations, our assessments on potential effects of warming and adaptation on RUE are only valid for a two-species ecosystem including a cold-water dinoflagellate of the genus *Apocalathium* and a cold-water
diatom of the genus *Thalassiosira*. For this species configuration, we found that adaptation increases resource use efficiency under both control and warming conditions. For both climate scenarios, adaptation is driven by competition for nitrogen, allowing *Apocalathium* and *Thalassiosira* to use the available nitrogen optimally within their means.

*Apocalathium* can only grow within a specific temperature niche, with the freezing point
of sea water at the lower end and the fixed temperature threshold of encystment (6 °C) at the upper end (see Hinners et al., 2019 and Sect. 4.1). Within this fixed niche, *Apocalathium* adapts to lower temperatures under control conditions due to intraspecific competition for nitrogen. Since nitrogen concentration is highest during the initial phase of the bloom (Fig. B5), the environment selects for early bloomers with comparatively low optimum temperatures, which
grow first and leave less nitrogen for individuals with higher optimum temperatures. Selection for early bloomers advances the bloom peak by more than 1 week compared to the control scenario without adaptation, which extends the bloom duration by a few days. Bloom duration, in this context, refers to the time during which growing stages reach a minimum concentration of 0.05 mmol N m$^{-3}$.

On the contrary to *Apocalathium*, *Thalassiosira* is not restricted by its life cycle and can therefore occupy its optimal niche more flexibly. Under control conditions, *Thalassiosira* adapts to higher temperatures to (I) delay its bloom by ~18 d to reduce competition with *Apocalathium,* and (II) merge its spring and autumn blooms into a single bloom, which persists from June until December (Fig. B5). In this way, *Thalassiosira* can continuously take up
nitrogen for 5 months in a row until light becomes limiting in winter.

When temperatures increase under global warming, RUE decreases but remains at a higher level when adaptation is enabled. Without adaptation, the spring bloom of *Apocalathium* is shifted by ~7 d towards winter, with the peak amplitude decreasing by ~18 % (Fig. B5). These warming-related changes lead to a decrease in bloom duration of more than 2 weeks. The spring
and autumn blooms of *Thalassiosira* are shifted towards winter as well, and even more than the spring bloom of *Apocalathium* (~26 d and ~24 d, respectively). However, both *Thalassiosira* blooms only show minor changes in peak amplitude and duration.



With adaptation, *Thalassiosira* does not show notable warming-related changes in bloom timing, duration, or amplitude. The spring bloom of *Apocalathium*, on the contrary, is again by more than 2 weeks shorter, meaning that the shortening is not caused by lacking adaptation but by the fixed temperature threshold of encystment. Still, with adaptation, we observe a slightly smaller shift in bloom timing of *Apocalathium* with ~5 d instead of ~7 d, and a ~16 % higher peak amplitude.

To conclude, our simulations show that adaptation generally allows for a more efficient use of resources and thus higher RUE. Models that ignore adaptation may hence systematically underestimate RUE under both present-day and future conditions. However, our projected warming-related decrease in RUE only applies to the species configuration in our model. We cannot make statements about future changes in RUE in other ecosystems with a different set of species. Future work can build on our results and investigate RUE in more complex ecosystems to make more general statements on future warming-related changes. Our results demonstrate that future models should consider not only adaptation, but also possible species-specific constraints on adaptation, such as life cycle dynamics.

### 4.6 Control factors and feedbacks in our model ecosystem

We found that all ecosystem functions are positively correlated in our simulations, with dinoflagellate annual biomass being the only exception. Under control conditions, all correlations (regardless of their direction) are stronger with phytoplankton adaptation, when niche separation allows for a stronger cyanobacterial summer bloom (see Sect. 3.1). Due to the stronger cyanobacterial summer bloom, more atmospheric nitrogen is fixed. The increase in nitrogen fixation is beneficial especially for diatoms, which can directly take up the newly available nitrogen in autumn. Dinoflagellates, however, do not benefit from increased nitrogen fixation. During spring bloom, dinoflagellates reach a higher maximum concentration than diatoms. Since we assume that zooplankton grazing depends on phytoplankton biomass only (see Sect. 4.2), grazing is stronger on dinoflagellates than on diatoms. Indeed, zooplankton peak during dinoflagellate spring bloom, meaning that dinoflagellates constitute the main food source for zooplankton. Hence, the stronger dinoflagellates grow due to increased nitrogen fixation, the more they are grazed by zooplankton, and increased nitrogen fixation has no positive impact on dinoflagellate biomass. Zooplankton, on the contrary, benefit indirectly from increased nitrogen fixation. To conclude, adaptation induces positive feedback in our control simulations: Dinoflagellates and diatoms adapt to individual temperature niches to reduce



competition for nitrogen, with the reduced competition between diatoms and cyanobacteria allowing for a stronger cyanobacterial summer bloom. While the increased cyanobacterial nitrogen fixation has a direct positive effect on diatoms, zooplankton benefit indirectly through stronger grazing on dinoflagellates. The result is an overall increase in biomass production, which, in turn, increases carbon export.

Under global warming, we observe a similar positive feedback mechanism for W, where phytoplankton adaptation is disabled. For WA, on the contrary, we find an overall weakening of correlations, even if cyanobacteria are stronger in WA than in CA. Correlations in WA are weaker especially for dinoflagellates and zooplankton, with the negative correlation between them turning slightly positive. Correlations for diatoms are weakened as well but to a lesser extent. Due to the stronger cyanobacterial summer bloom, nitrogen fixation increases in WA compared to CA, which is again beneficial for diatoms. As a result, grazing pressure on diatoms increases and weakens the positive correlation between diatoms and zooplankton. In addition to the enhanced grazing pressure, there is another factor that reduces the benefit of the increasing cyanobacteria for diatoms. As demonstrated by Hochfeld & Hinners (2024), cyanobacteria restrict diatom adaptation to the increasing temperatures in WA due to their presence in summer, leading to a stronger shift of the two diatom blooms towards winter. While this is not necessarily problematic for the diatom autumn bloom if sufficient light is available, it is for the spring bloom since dinoflagellates are present at lower temperatures. Thus, we observe a weaker positive correlation between diatoms and cyanobacteria in WA than in CA. Due to the stronger grazing on diatoms, zooplankton are also less positively impacted by cyanobacteria. The weaker positive effect of cyanobacteria on diatoms and zooplankton is reflected in a slight weakening of the remaining positive correlations, and a notable weakening of the negative correlations with dinoflagellates. Furthermore, the reduced relative grazing pressure on dinoflagellates reverses the negative correlation with zooplankton, meaning that an increase in zooplankton biomass no longer implies a decrease in dinoflagellate biomass.

To conclude, cyanobacteria are the most important control factor in our model ecosystem, which is also confirmed by a principal component analysis (Fig. B6). First, cyanobacteria produce the highest amount of biomass per year. Second, due to their ability to fix atmospheric nitrogen, they directly control the biomass production of dinoflagellates and diatoms, and indirectly of zooplankton. Cyanobacteria are therefore the main factor for carbon export in our simulations, which also agrees with observations as discussed above (see Sect. 4.3). However, the interdependencies between cyanobacteria and the other taxa may change



depending on the climate scenario and the presence or absence of phytoplankton adaptation. Under control conditions and in W, there are clear losers and winners of increased nitrogen fixation among the phytoplankton, with dinoflagellates being the losers and diatoms being the winners. In WA, these dynamics begin to reverse slightly since cyanobacteria restrict diatoms in their adaptation to higher temperatures. These results demonstrate that by neglecting adaptation, we may be missing adaptation-related changes in taxa interactions, especially in changing environments, which can affect the entire ecosystem and hence its functioning.

## 5 Conclusions

Our study demonstrates that phytoplankton adaptation does not only affect simulated phytoplankton dynamics themselves but also simulated ecosystem functions through bottom-up control. The effect of phytoplankton adaptation on simulated ecosystem functions depends on environmental conditions.

In a steady environment, phytoplankton adaptation allows for a more efficient use of resources through niche separation, which, in turn, increases primary production. An increase in primary production may enhance secondary production, nitrogen fixation, and carbon export, and maybe even other ecosystem functions not included in this study. Thus, by neglecting adaptation, models can systematically underestimate resource use efficiency in a steady environment and hence ecosystem functions that are directly related to primary production. In a warming environment, however, adaptation has the opposite effect. With the ability to adapt to the increasing temperatures, non-pre-adapted taxa can mitigate the dominance of superior pre-adapted taxa. Since different taxa fulfill different functions in the ecosystem, weaker changes in their abundance lead to weaker changes in related ecosystem functions. By neglecting phytoplankton adaptation, models may therefore systematically overestimate warming-related changes in ecosystem functioning. To realistically simulate ecosystem functioning in both steady and changing environments, future models should not only consider multiple phytoplankton functional groups due to their different roles in the ecosystem but also their potential to adapt to their environment. Our study furthermore suggests that models without adaptation may miss adaptation-related interdependencies between taxa that may play out differently in steady and changing environments and can hence lead to changes in ecosystem dynamics and functioning. In addition, our study highlights the importance of life cycle



dynamics for phytoplankton responses to global warming due to potential feedback mechanisms and/or adaptation constraints.

Our study is a first step to improve model projections of future ecosystem-level changes. Future work can build on our results, for example by expanding on our model ecosystem to include multiple nutrients, a higher diversity of phytoplankton functional groups, a more

complex representation of zooplankton, and higher trophic levels. Another next step would be to couple our or a similar evolutionary ecosystem model to a 1D or 3D physical environment to allow for a more realistic representation of physically driven processes, e.g., biogeochemical cycling. The performance of such an evolutionary biogeochemical-physical model could then be tested against long-term evolutionary data (e.g., from sediment archives). Using such a

validated model for climate projections could notably improve estimations of future ecosystem-level changes.



# Appendices

## Appendix A

Statistical *t*-test results for the model output presented in Table 2 (Sect. 3.2) and Table 3 (Sect. 3.3).

**Table A1:** Results of a series of *t*-tests comparing all model scenarios (C: control, CA: control and adaptation, W: warming, WA: warming and adaptation) with regard to zooplankton bloom timing,
zooplankton peak abundance, and the time lag between the peaks of zooplankton and phytoplankton. The table presents the value of the test statistic (*t*), the degrees of freedom (df), and the *p*-value (*p*). Please note that we used a paired-sample *t*-test when comparing control and warming simulations since these were performed pairwise, and a two-sample *t*-test otherwise.

|  | Variable | *t* | df | *p* |
|---|---|---|---|---|
| CA vs. C | Timing | 13.2463 | 12 | $1.5965 \times 10^{-8}$ |
|  | Abundance | -6.9046 | 12 | $1.6404 \times 10^{-5}$ |
|  | Time lag | 6.0295 | 6 | $9.4005 \times 10^{-4}$ |
| WA vs. W | Timing | 8.1747 | 12 | $3.0117 \times 10^{-6}$ |
|  | Abundance | 2.4289 | 12 | 0.0318 |
|  | Time lag | -3.4739 | 6 | 0.0132 |
| W vs. C | Timing | 27.9240 | 6 | $1.3954 \times 10^{-7}$ |
|  | Abundance | -31.5978 | 6 | $6.6762 \times 10^{-8}$ |
|  | Time lag | 16.2498 | 6 | $3.4561 \times 10^{-6}$ |
| WA vs. CA | Timing | 7.3860 | 6 | $3.1602 \times 10^{-4}$ |
|  | Abundance | -4.6286 | 6 | 0.0036 |
|  | Time lag | 3.9232 | 6 | 0.0078 |

**Table A2:** Results of a series of *t*-tests comparing all model scenarios (C: control, CA: control and adaptation, W: warming, WA: warming and adaptation) with regard to annual balances. The table presents the value of the test statistic (*t*), the degrees of freedom (df), and the *p*-value (*p*). Please note that we used a paired-sample *t*-test when comparing control and warming simulations since these were performed pairwise, and a two-sample *t*-test otherwise.



| | Variable | $t$ | df | $p$ |
|---|---|---|---|---|
| **CA vs. C** | Dinoflagellates | 2.1795 | 12 | 0.0499 |
| | Diatoms | -7.7662 | 12 | $5.0873 \times 10^{-6}$ |
| | Cyanobacteria | -6.1108 | 12 | $5.2491 \times 10^{-5}$ |
| | Phytoplankton | -6.4065 | 12 | $3.3697 \times 10^{-5}$ |
| | Zooplankton | -9.1802 | 12 | $8.9508 \times 10^{7}$ |
| | $N_2$ fixation | -5.8068 | 12 | $8.3836 \times 10^{-5}$ |
| | Carbon export | -5.8882 | 12 | $7.3861 \times 10^{-5}$ |
| | RUE | -27.2736 | 12 | $3.6372 \times 10^{-12}$ |
| **WA vs. W** | Dinoflagellates | -1.9463 | 12 | 0.0754 |
| | Diatoms | -3.0493 | 12 | 0.0101 |
| | Cyanobacteria | -13.7101 | 12 | $1.0818 \times 10^{-8}$ |
| | Phytoplankton | -12.5522 | 12 | $2.9249 \times 10^{-8}$ |
| | Zooplankton | -7.3374 | 12 | $9.0067 \times 10^{-6}$ |
| | $N_2$ fixation | -12.1507 | 12 | $4.2078 \times 10^{-8}$ |
| | Carbon export | -12.8997 | 12 | $2.1524 \times 10^{-8}$ |
| | RUE | 25.0575 | 12 | $9.8930 \times 10^{-12}$ |
| **W vs. C** | Dinoflagellates | 18.1062 | 6 | $1.8266 \times 10^{-6}$ |
| | Diatoms | -31.8063 | 6 | $6.4192 \times 10^{-8}$ |
| | Cyanobacteria | -99.4698 | 6 | $6.9577 \times 10^{-11}$ |
| | Phytoplankton | -77.4443 | 6 | $3.1205 \times 10^{-10}$ |
| | Zooplankton | -39.0206 | 6 | $1.8926 \times 10^{-8}$ |
| | $N_2$ fixation | -88.6053 | 6 | $1.3921 \times 10^{-10}$ |
| | Carbon export | -77.9701 | 6 | $2.9965 \times 10^{-10}$ |
| | RUE | 48.8723 | 6 | $4.9211 \times 10^{-9}$ |
| **WA vs. CA** | Dinoflagellates | 9.4959 | 6 | $7.7730 \times 10^{-5}$ |
| | Diatoms | -12.3243 | 6 | $1.7400 \times 10^{-5}$ |
| | Cyanobacteria | -8.9350 | 6 | $1.0966 \times 10^{-4}$ |
| | Phytoplankton | -9.0959 | 6 | $9.9165 \times 10^{-5}$ |
| | Zooplankton | -5.2772 | 6 | 0.0019 |
| | $N_2$ fixation | -8.2710 | 6 | $1.6905 \times 10^{-4}$ |
| | Carbon export | -9.3836 | 6 | $8.3152 \times 10^{-5}$ |
| | RUE | 62.3327 | 6 | $1.1462 \times 10^{-9}$ |






## Appendix B

Supporting figures for Sects. 2.3, 3.2, 3.3, 4.1, 4.5, and 4.6.

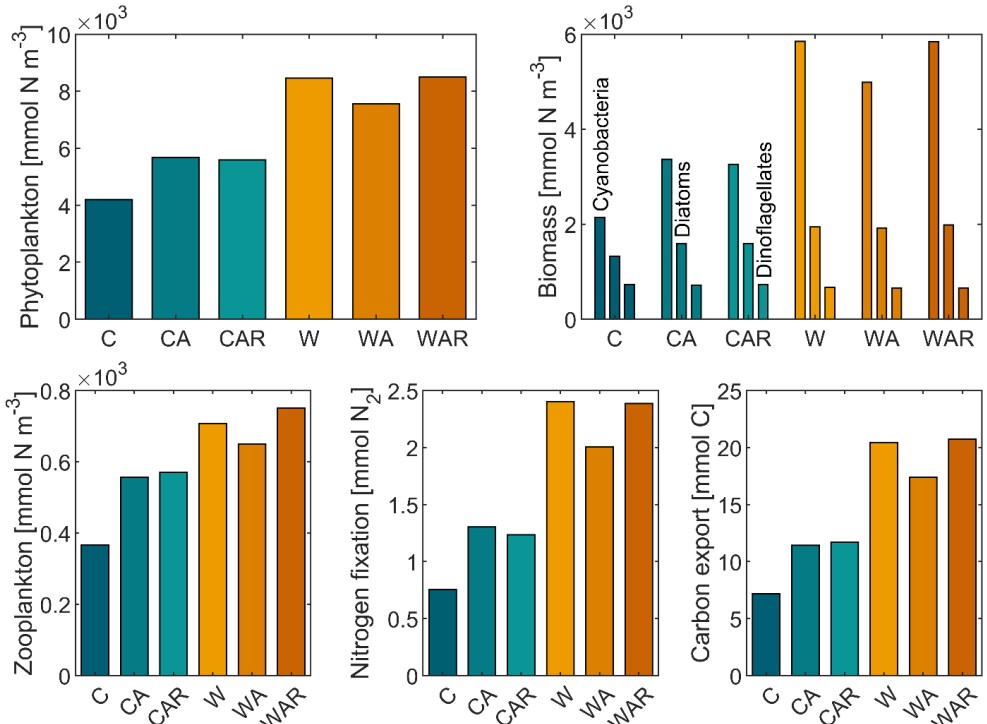

**Figure B1:** Annual balances for additional simulations with resuspension (C: control, CA: control and adaptation, CAR: control, adaptation, and resuspension, W: warming, WA: warming and adaptation, WAR: warming, adaptation, and resuspension). Carbon export is corrected for the carbon content of resuspended resting cells. Please note that we excluded resource use efficiency (RUE) from the figure since RUE simulations with resuspension are not comparable with those without.





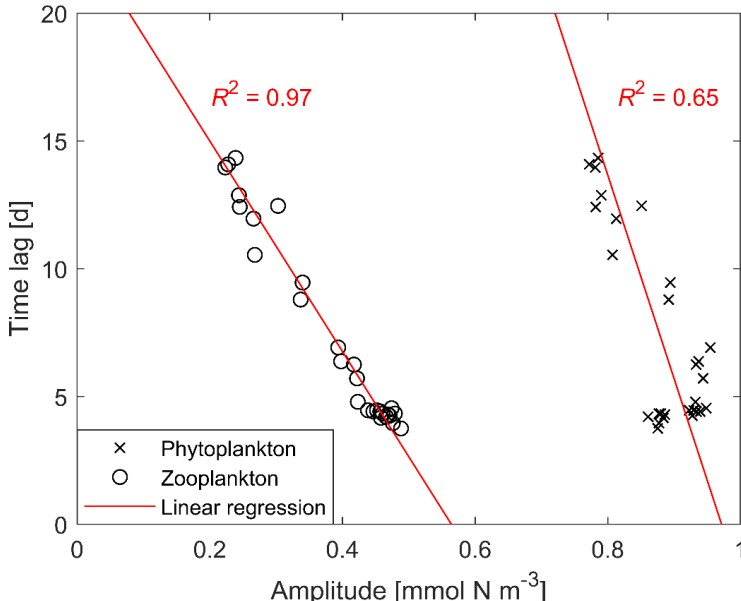

**Figure B2:** Time lag between phytoplankton and zooplankton blooms as a function of the peak amplitudes of phytoplankton and zooplankton, respectively. Shown are the time lags for the last simulation year of seven different simulations per model scenario, including linear regressions with both phytoplankton and zooplankton peak amplitudes and the corresponding $R^2$-values.



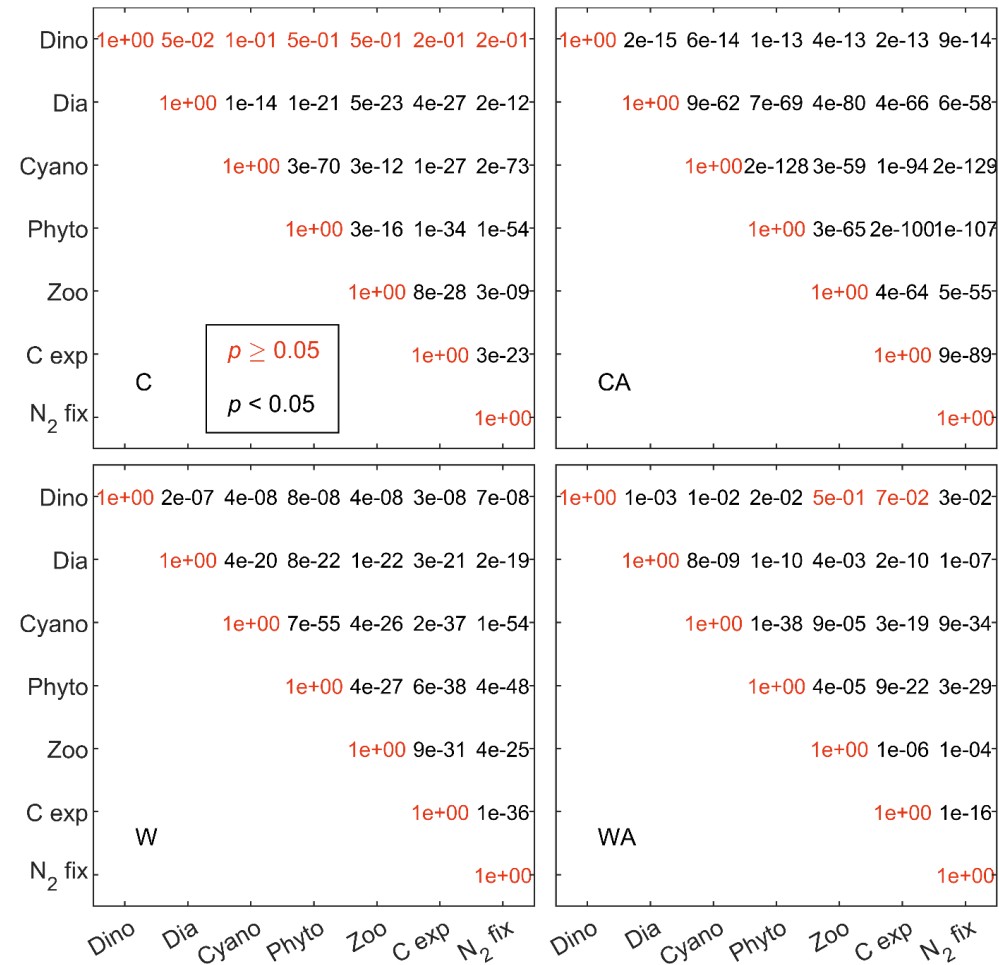


**Figure B3:** Matrices showing the *p*-values for the correlations in Fig. 5 (Sect. 3.3). Model scenario abbreviations: C: control, CA: control and adaptation, W: warming, WA: warming and adaptation. Black numbers indicate that the corresponding correlations are statistically significant at the 0.05 level, while orange numbers indicate the opposite.






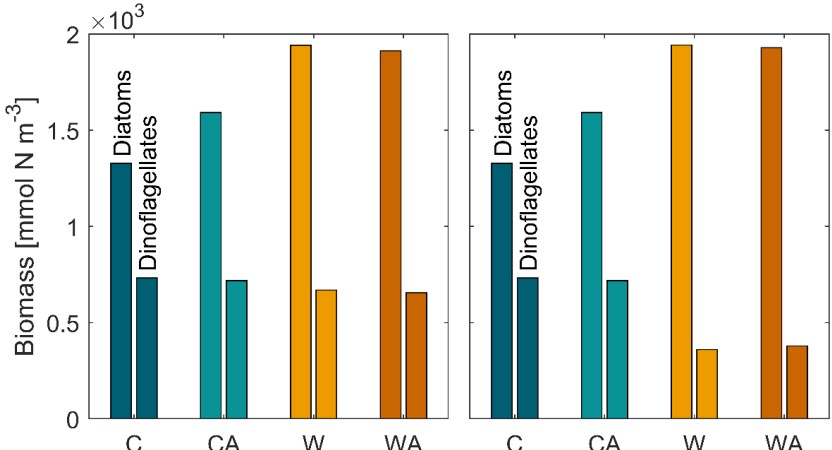

**Figure B4:** Annual biomasses of the cold-water dinoflagellate *Apocalathium* and the cold-water diatom *Thalassiosira* for the four different model scenarios (C: control, CA: control and adaptation, W: warming, WA: warming and adaptation). Left: Results for our standard simulations with a fixed encystemt rate of *Apocalathium*. Right: Results for additional simulations, in which we artificially decreased the encystment rate of *Apocalathium* at the rate found by Hinners et al (2017).


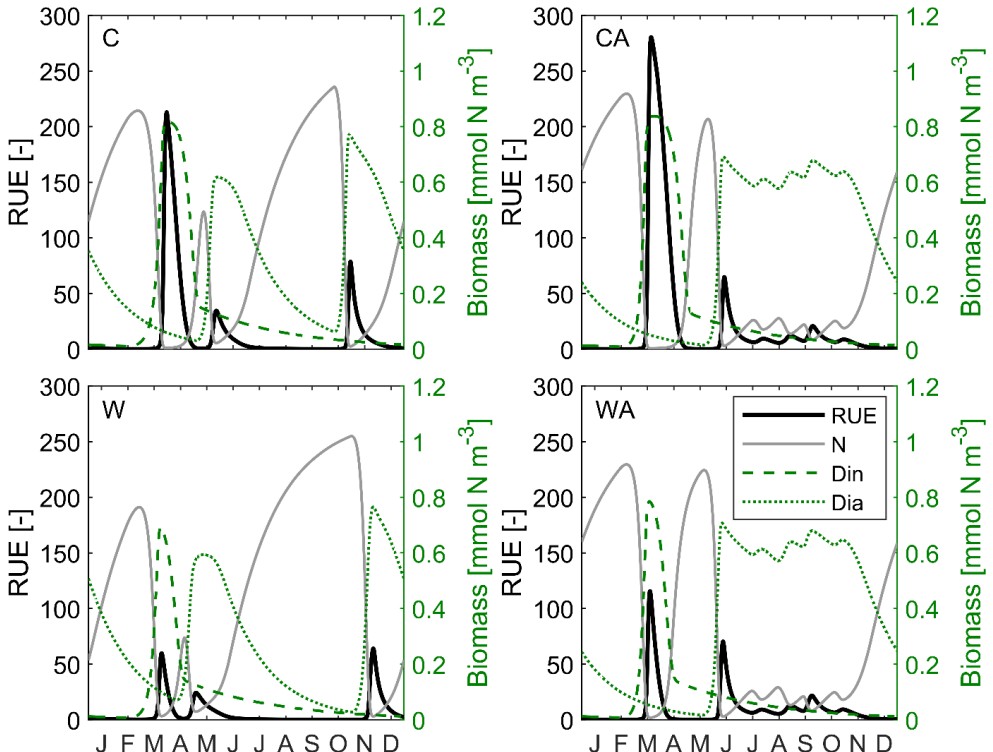

**Figure B5:** Resource use efficiency (RUE) of our focal dinoflagellate and diatom species of the genera
*Apocalathium* and *Thalassiosira* throughout the seasonal cycle for all model scenarios (C: control, CA:
control and adaptation, W: warming, WA: warming and adaptation). The figure shows results for the last
simulation year, which were averaged over seven different simulations per scenario. Also shown are the
nitrogen concentration (N), and the biomasses of *Apocalathium* (Din) and *Thalassiosira* (Dia).





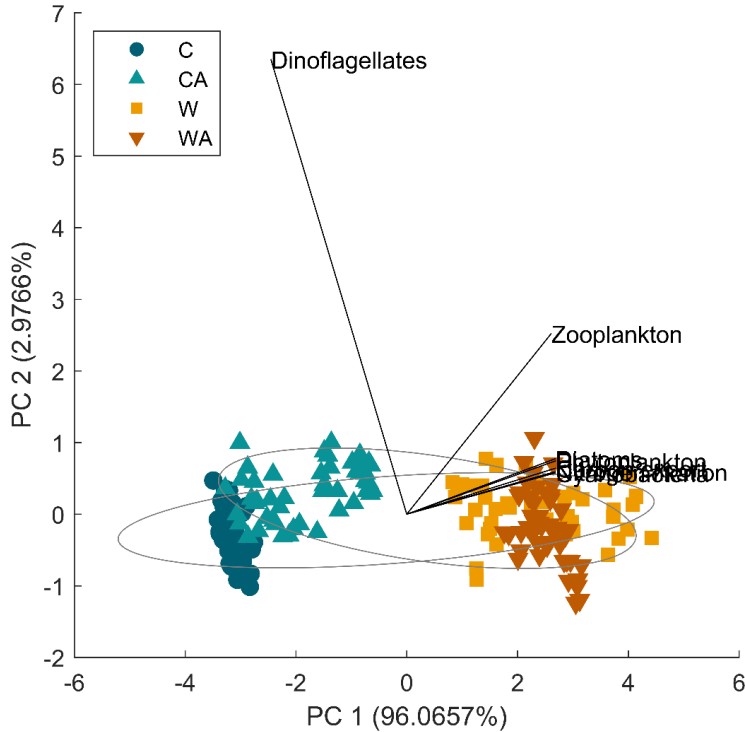

**Figure B6:** Results for a principal component analysis (PCA). The PCA shows that most variability in our model ecosystem can be explained by the first principal component (PC 1), which is associated with all model variables that are positively impacted by cyanobacteria. Zooplankton and especially dinoflagellates can be clearly identified as outliers.

# Code availability

The model code and the scripts for evaluating the model output and creating the figures are available on GitHub at https://github.com/Isabell-Hochfeld/Adaptive-Phytoplankton-Community-Model, last access: April 23, 2024) and on Zenodo at https://zenodo.org/doi/10.5281/zenodo.10693812 (version 1.1.0, Hochfeld, 2024). All code is written in MATLAB (version R2022a).

# Author contribution

JH and IH designed the study. IH modified the model, performed the model simulations, and analyzed the model output. Both authors contributed to writing the manuscript.



## Competing Interests

The authors declare that they have no conflict of interest.

## Acknowledgements

Funding was provided through the project PhytoArk (K314/2020) funded by the Leibniz Association.

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
