# Peer review of "Phytoplankton adaptation to steady or changing environments affects marine ecosystem functioning"

_EGUsphere, 2024_

## Referee Comment (RC2)

Hochfeld and Hinners evaluate the effect of phytoplankton adaptation to ecosystem functioning by using a 0-dimensional evolutionary ecosystem model. They used a previously implemented individual-based model from the same authors with the following components: one nutrient (nitrogen), three phytoplankton functional groups (cyanobacteria, dinoflagellates and diatoms), one generic zooplankton and a detritus pool. Phytoplankton adaptation was implemented as two 'flexible traits' as they called them, namely optimum temperature for growth and cell size, where the variation of the former is affected by a random mutation process and the latter by cell growth. They conclude by highlighting the importance of including phytoplankton adaptation in ecosystem models.

First, it is difficult to assess the scientific significance of their modelling approach. The model is briefly presented in this manuscript but is published in the supplementary material of a previous article by the same authors (Hochfeld and Hinners, 2024). The authors repeatedly refer to this work to describe the model, without explicitly explaining what they have 'slightly modified' or 'extended'. In particular, it is unclear what are the assumptions, processes, and parameters of the model, how they quantified them in their current model description and how these differ from their previous manuscript. Perhaps the individual-based modelling approach used by the authors is novel, albeit very hard to judge based on the current description.

Second, they used the same numerical experiments as in the previous work, which show the same results as in their previous ms (cf. figure 2 in this and their previous ms with biomass staked vs non stacked). Thus, the only new results seem to be the quantification of the ecosystem functions (later results in the current manuscript), but the conclusion between manuscripts is nearly the same. Overall, I would have appreciated a thorough presentation of the model, with its assumptions, and limitations, to better interpret how these impact the author's interpretations and how much their current analysis advances our understanding of adaptive response of phytoplankton and its impact to ecological functions. Unfortunately, I cannot see that in the current manuscript and less so in comparison to their previous work.

Third, competition for resources and phytoplankton adaptation based on functional traits related to size and to a lesser extent to temperature has been investigated in models with different complexity, covering various scales and with various modelling approaches (e.g.: Bruggeman and Kooijman, 2007; Follows et al., 2007; Hellweger and Kianirad, 2007; Pahlow et al., 2008; Merico et al., 2009; Banas, 2011; Clark et al., 2011; Norberg et al., 2012; Thomas et al., 2012; Ward et al., 2012; Toseland et al., 2013; Wirtz, 2013; Daines et al., 2014; Terseleer et al., 2014; Smith et al., 2015; Kerimoglu et al., 2017; Kremer et al., 2017a; Taherzadeh et al., 2017; Acevedo-Trejos et al., 2018; Chen et al., 2019; Dutkiewicz et al., 2020). Some of these eco-evolutionary trait-based modelling approaches have been reviewed over the past decades (Norberg, 2004; Anderson, 2005, 2010; Litchman and Klausmeier, 2008; Hellweger and Bucci, 2009; Smith et al., 2011; Follows and Dutkiewicz, 2011; Andersen et al., 2015; Bonachela et al., 2016; Hellweger et al., 2016; Kremer et al., 2017b; Ward et al., 2019; Zakharova et al., 2019; Kiørboe and Andersen, 2019; Klausmeier et al., 2020). However, the introduction only covers a few examples and gives the impression that not much work has been done in the past decades to capture the adaptive capacity of planktonic organisms in ecosystem models, which to my knowledge is not the case. Hence, I consider that the introduction needs to provide a better rationale for the study in the context of previous ecoevolutionary trait-based models, what technical or knowledge gap is covered and to clearly present what is distinct in their modelling approach.

Last, in both manuscripts the authors suggest that their model aims to capture the dynamics of the Baltic Sea. However, no model calibration or validation against observations is provided. If the authors want to make such a claim, I would suggest having figures that clearly show model performance against observations.

Albeit the presentation quality of the manuscript is good, unfortunately, the various issues I have listed above do not allow me to recommend the manuscript for publication in it is current state.

References

Acevedo-Trejos, E., Marañón, E., and Merico, A. (2018). Phytoplankton size diversity and ecosystem function relationships across oceanic regions. *Proc. R. Soc. B Biol. Sci.* 285. doi:10.1098/rspb.2018.0621.

Andersen, K. H., Berge, T., Gonçalves, R. J., Hartvig, M., Heuschele, J., Hylander, S., et al. (2015). Characteristic sizes of life in the oceans, from bacteria to whales. *Ann. Rev. Mar. Sci.* 8, 1–25. doi:10.1146/annurev-marine-122414-034144.

Anderson, T. R. (2005). Plankton functional type modelling: running before we can walk? *J. Plankton Res.* 27. doi:10.1093/plankt/fbi076.

Anderson, T. R. (2010). Progress in marine ecosystem modelling and the "unreasonable effectiveness of mathematics". *J. Mar. Syst.* 81, 4–11. doi:10.1016/j.jmarsys.2009.12.015.

Banas, N. S. (2011). Adding complex trophic interactions to a size-spectral plankton model: Emergent diversity patterns and limits on predictability. *Ecol. Modell.* 222, 2663–2675. doi:10.1016/j.ecolmodel.2011.05.018.

Bonachela, J. A., Klausmeier, C. A., Edwards, K. F., Litchman, E., and Levin, S. A. (2016). The role of phytoplankton diversity in the emergent oceanic stoichiometry. *J. Plankton Res.* 38, 1021–1035. doi:10.1093/plankt/fbv087.

Bruggeman, J., and Kooijman, S. A. L. M. (2007). A biodiversity-inspired approach to aquatic ecosystem modeling. *Limnol. Oceanogr.* 52, 1533–1544. doi:10.4319/lo.2007.52.4.1533.

Chen, B., Smith, S. L., and Wirtz, K. W. (2019). Effect of phytoplankton size diversity on primary productivity in the North Pacific: trait distributions under environmental variability. *Ecol. Lett.* 22, 56–66. doi:10.1111/ele.13167.

Clark, J. R., Daines, S. J., Lenton, T. M., Watson, A. J., and Williams, H. T. P. (2011). Individual-based modelling of adaptation in marine microbial populations using genetically defined physiological parameters. *Ecol. Modell.* 222, 3823–3837. doi:10.1016/j.ecolmodel.2011.10.001.

Daines, S. J., Clark, J. R., and Lenton, T. M. (2014). Multiple environmental controls on phytoplankton growth strategies determine adaptive responses of the N:P ratio. *Ecol. Lett.* 17, 414–425. doi:10.1111/ele.12239.

Dutkiewicz, S., Cermeno, P., Jahn, O., Follows, M. J., Hickman, A. E., Taniguchi, D. A. A., et al. (2020). Dimensions of marine phytoplankton diversity. *Biogeosciences* 17, 609–634. doi:10.5194/bg-17-609-2020.

Follows, M. J., and Dutkiewicz, S. (2011). Modeling diverse communities of marine microbes. *Ann. Rev. Mar. Sci.* 3, 427–451. doi:10.1146/annurev-marine-120709-142848.

Follows, M. J., Dutkiewicz, S., Grant, S., and Chisholm, S. W. (2007). Emergent biogeography of microbial communities in a model ocean. *Science.* 315, 1843–1846. doi:10.1126/science.1138544.

Hellweger, F. L., and Bucci, V. (2009). A bunch of tiny individuals-Individual-based modeling for microbes. *Ecol. Modell.* 220, 8–22. doi:10.1016/j.ecolmodel.2008.09.004.

Hellweger, F. L., Clegg, R. J., Clark, J. R., Plugge, C. M., and Kreft, J. U. (2016). Advancing microbial sciences by individual-based modelling. *Nat. Rev. Microbiol.* 14, 461–471. doi:10.1038/nrmicro.2016.62.

Hellweger, F. L., and Kianirad, E. (2007). Individual-based modeling of phytoplankton: Evaluating approaches for applying the cell quota model. *J. Theor. Biol.* 249, 554–565. doi:10.1016/j.jtbi.2007.08.020.

Hochfeld, I., and Hinners, J. (2024). Evolutionary adaptation to steady or changing environments affects competitive outcomes in marine phytoplankton. *Limnol. Oceanogr.* 69, 1172–1186. doi:10.1002/lno.12559.

Kerimoglu, O., Hofmeister, R., Maerz, J., Riethmüller, R., and Wirtz, K. W. (2017). The acclimative biogeochemical model of the southern North Sea. *Biogeosciences* 14, 4499–4531. doi:10.5194/bg-14-4499-2017.

Kiørboe, T., and Andersen, K. H. (2019). Nutrient affinity, half-saturation constants and the cost of toxin production in dinoflagellates. *Ecol. Lett.* 22, 558–560. doi:10.1111/ele.13208.

Klausmeier, C. A., Kremer, C. T., and Koffel, T. (2020). 'Trait-based ecological and eco-evolutionary theory', in *Theoretical Ecology: concepts and applications*, eds. K. S. McCann and G. Gellner (Oxford University Press).

Kremer, C. T., Thomas, M. K., and Litchman, E. (2017a). Temperature- and size-scaling of phytoplankton population growth rates: reconciling the Eppley curve and the metabolic theory of ecology. *Limnol. Oceanogr.* doi:10.1002/lno.10523.

Kremer, C. T., Williams, A. K., Finiguerra, M., Fong, A. A., Kellerman, A., Paver, S. F., et al. (2017b). Realizing the potential of trait-based aquatic ecology: New tools and collaborative approaches. *Limnol. Oceanogr.* 62, 253–271. doi:10.1002/lno.10392.

Litchman, E., and Klausmeier, C. A. (2008). Trait-based community ecology of phytoplankton. *Annu. Rev. Ecol. Evol. Syst.* 39, 615–639. doi:10.1146/annurev.ecolsys.39.110707.173549.

Merico, A., Bruggeman, J., and Wirtz, K. (2009). A trait-based approach for downscaling complexity in plankton ecosystem models. *Ecol. Modell.* 220, 3001–3010. doi:10.1016/j.ecolmodel.2009.05.005.

Norberg, J. (2004). Biodiversity and ecosystem functioning: A complex adaptive systems approach. *Limnol. Oceanogr.* 49, 1269–1277. doi:10.4319/lo.2004.49.4_part_2.1269.

Norberg, J., Urban, M. C., Vellend, M., Klausmeier, C. a., and Loeuille, N. (2012). Eco-evolutionary responses of biodiversity to climate change. *Nat. Clim. Chang.* 2, 747–751. doi:10.1038/nclimate1588.

Pahlow, M., Vézina, A. F., Casault, B., Maass, H., Malloch, L., Wright, D. G., et al. (2008). Adaptive model of plankton dynamics for the North Atlantic. *Prog. Oceanogr.* 76, 151–191. doi:10.1016/j.pocean.2007.11.001.

Smith, S. L., Pahlow, M., Merico, A., Acevedo-Trejos, E., Sasai, Y., Yoshikawa, C., et al. (2015). Flexible phytoplankton functional type (FlexPFT) model: size-scaling of traits and

optimal growth. *J. Plankton Res.* 38, 977–992. doi:10.1093/plankt/fbv038.

Smith, S. L., Pahlow, M., Merico, A., and Wirtz, K. W. (2011). Optimality-based modeling of planktonic organisms. *Limnol. Oceanogr.* 56, 2080–2094. doi:10.4319/lo.2011.56.6.2080.

Taherzadeh, N., Kerimoglu, O., and Wirtz, K. W. (2017). Can we predict phytoplankton community size structure using size scalings of eco-physiological traits? *Ecol. Modell.* 360, 279–289. doi:10.1016/j.ecolmodel.2017.07.008.

Terseleer, N., Bruggeman, J., Lancelot, C., and Gypens, N. (2014). Trait-based representation of diatom functional diversity in a plankton functional type model of the eutrophied Southern North Sea. *Limnol. Oceanogr.* 59, 1–16. doi:10.4319/lo.2014.59.6.0000.

Thomas, M. K., Kremer, C. T., Klausmeier, C. A., and Litchman, E. (2012). A global pattern of thermal adaptation in marine phytoplankton. *Science.* 338, 1085–8. doi:10.1126/science.1224836.

Toseland, A., Daines, S. J., Clark, J. R., Kirkham, A., Strauss, J., Uhlig, C., et al. (2013). The impact of temperature on marine phytoplankton resource allocation and metabolism. *Nat. Clim. Chang.* 3, 1–6. doi:10.1038/nclimate1989.

Ward, B. A., Collins, S., Dutkiewicz, S., Gibbs, S., Bown, P., Ridgwell, A., et al. (2019). Considering the Role of Adaptive Evolution in Models of the Ocean and Climate System. *J. Adv. Model. Earth Syst.* 11, 3343–3361. doi:10.1029/2018MS001452.

Ward, B. A., Dutkiewicz, S., Jahn, O., and Follows, M. J. (2012). A size-structured food-web model for the global ocean. *Limnol. Oceanogr.* 57, 1877–1891. doi:10.4319/lo.2012.57.6.1877.

Wirtz, K. W. (2013). Mechanistic origins of variability in phytoplankton dynamics: Part I: niche formation revealed by a size-based model. *Mar. Biol.* 160, 2319–2335. doi:10.1007/s00227-012-2163-7.

Zakharova, L., Meyer, K. M., and Seifan, M. (2019). Trait-based modelling in ecology: A review of two decades of research. *Ecol. Modell.* 407, 108703. doi:10.1016/j.ecolmodel.2019.05.008.

---

## Author Response (AR1)

**Author's response**

**1. Comments from the referees**

Reviewer 1:

This article builds on previous work by the same authors, exploring how the representation of phytoplankton evolution can affect the predictions of a plankton ecosystem model under stable and warming conditions. In a previous article (Hochfield & Hinners 2024), the authors explored a very similar model under a similar set of environmental scenarios. The article demonstrated that allowing phytoplankton to adapt to changing environmental conditions impacted projections of community composition. In the new manuscript the authors extend this analysis to include the effects of evolution on ecosystem function.

While there is significant overlap between the two articles, there are new results in the new manuscript. It is important however that the authors make a clearer distinction between the two manuscripts, explicitly noting what is repeated, what is different in terms of model setup and assumptions, and what new findings where not seen previously. It appears that the main distinction between the papers is in the additional focus on ecosystem functions (i.e. carbon export, nitrogen fixation and resource use efficiency). That said, much of the Results and Discussion sections still focus on community composition, and there is a possibility of some duplication. I would recommend that these sections are rewritten with clearer reference to the previous work, stating what was shown before and what has been added in the new manuscript. As an example, the authors note on line 126 that the original model has been "slightly extended", but it was not immediately clear to me what the differences were.

In addition to this issue, I found that the findings of the paper do not appear to be generally informative on how evolving and non-evolving plankton ecosystems might respond to climate change. Rather, we have a lot of information on how this particular configuration of this particular model responds under a very specific set of environmental forcings. Correlations and changes are presented as a long list, with little context of why the modelled changes should be of interest. It is quite difficult as a reader to understand the relevance of all these details.

I would urge the authors to develop a clearer narrative structure to the manuscript, identifying early on a small set of robust findings they want to communicate. The results can then be presented in a way that supports of refutes these ideas. Being more specific, the Results section provides a list of very precise quantitative results, but it is hard to know what these mean in a broader context. Why do we need to know how each individual model species responds in each individual experiment? This is an important issue, because I suspect that the very detailed results will be quite sensitive to the model structure and the environmental scenario. To give one example, the projected climate change scenarios only included changes in environmental temperature and neglect any long-term changes in other physical factors such as stratification/mixing. It is important to note that projected increases in productivity

and biomass under future climate change might not be repeated if the model were to represent future decreases in nutrient supply in a more stratified system.

In summary, I would urge the authors to work on the paper's narrative structure, emphasising a small number of key general findings, and building the manuscript around those.

References

Hochfeld, I. and Hinners, J. (2024), Evolutionary adaptation to steady or changing environments affects competitive outcomes in marine phytoplankton. Limnol Oceanogr, 69: 1172-1186. https://doi.org/10.1002/lno.12559

Reviewer 2:

Hochfeld and Hinners evaluate the effect of phytoplankton adaptation to ecosystem functioning by using a 0-dimensional evolutionary ecosystem model. They used a previously implemented individual-based model from the same authors with the following components: one nutrient (nitrogen), three phytoplankton functional groups (cyanobacteria, dinoflagellates and diatoms), one generic zooplankton and a detritus pool. Phytoplankton adaptation was implemented as two 'flexible traits' as they called them, namely optimum temperature for growth and cell size, where the variation of the former is affected by a random mutation process and the latter by cell growth. They conclude by highlighting the importance of including phytoplankton adaptation in ecosystem models.

First, it is difficult to assess the scientific significance of their modelling approach. The model is briefly presented in this manuscript but is published in the supplementary material of a previous article by the same authors (Hochfeld and Hinners, 2024). The authors repeatedly refer to this work to describe the model, without explicitly explaining what they have 'slightly modified' or 'extended'. In particular, it is unclear what are the assumptions, processes, and parameters of the model, how they quantified them in their current model description and how these differ from their previous manuscript. Perhaps the individual-based modelling approach used by the authors is novel, albeit very hard to judge based on the current description. Second, they used the same numerical experiments as in the previous work, which show the same results as in their previous ms (cf. figure 2 in this and their previous ms with biomass staked vs non stacked). Thus, the only new results seem to be the quantification of the ecosystem functions (later results in the current manuscript), but the conclusion between manuscripts is nearly the same. Overall, I would have appreciated a thorough presentation of the model, with its assumptions, and limitations, to better interpret how these impact the author's interpretations and how much their current analysis advances our understanding of adaptive response of phytoplankton and its impact to ecological functions. Unfortunately, I cannot see that in the current manuscript and less so in comparison to their previous work.

Third, competition for resources and phytoplankton adaptation based on functional traits related to size and to a lesser extent to temperature has been investigated in models with different complexity, covering various scales and with various modelling approaches (e.g.: Bruggeman and Kooijman, 2007; Follows et al., 2007; Hellweger and Kianirad, 2007; Pahlow

et al., 2008; Merico et al., 2009; Banas, 2011; Clark et al., 2011; Norberg et al., 2012; Thomas et al., 2012; Ward et al., 2012; Toseland et al., 2013; Wirtz, 2013; Daines et al., 2014; Terseleer et al., 2014; Smith et al., 2015; Kerimoglu et al., 2017; Kremer et al., 2017a; Taherzadeh et al., 2017; Acevedo-Trejos et al., 2018; Chen et al., 2019; Dutkiewicz et al., 2020). Some of these eco-evolutionary trait-based modelling approaches have been reviewed over the past decades (Norberg, 2004; Anderson, 2005, 2010; Litchman and Klausmeier, 2008; Hellweger and Bucci, 2009; Smith et al., 2011; Follows and Dutkiewicz, 2011; Andersen et al., 2015; Bonachela et al., 2016; Hellweger et al., 2016; Kremer et al., 2017b; Ward et al., 2019; Zakharova et al., 2019; Kiørboe and Andersen, 2019; Klausmeier et al., 2020). However, the introduction only covers a few examples and gives the impression that not much work has been done in the past decades to capture the adaptive capacity of planktonic organisms in ecosystem models, which to my knowledge is not the case. Hence, I consider that the introduction needs to provide a better rationale for the study in the context of previous eco- evolutionary trait-based models, what technical or knowledge gap is covered and to clearly present what is distinct in their modelling approach.

Last, in both manuscripts the authors suggest that their model aims to capture the dynamics of the Baltic Sea. However, no model calibration or validation against observations is provided. If the authors want to make such a claim, I would suggest having figures that clearly show model performance against observations. Albeit the presentation quality of the manuscript is good, unfortunately, the various issues I have listed above do not allow me to recommend the manuscript for publication in it is current state.

References

Acevedo-Trejos, E., Marañón, E., and Merico, A. (2018). Phytoplankton size diversity and ecosystem function relationships across oceanic regions. Proc. R. Soc. B Biol. Sci. 285. doi:10.1098/rspb.2018.0621.

Andersen, K. H., Berge, T., Gonçalves, R. J., Hartvig, M., Heuschele, J., Hylander, S., et al. (2015). Characteristic sizes of life in the oceans, from bacteria to whales. Ann. Rev. Mar. Sci. 8, 1–25. doi:10.1146/annurev-marine-122414-034144.

Anderson, T. R. (2005). Plankton functional type modelling: running before we can walk? J. Plankton Res. 27. doi:10.1093/plankt/qi076. Anderson, T. R. (2010). Progress in marine ecosystem modelling and the "unreasonable effectiveness of mathematics". J. Mar. Syst. 81, 4–11. doi:10.1016/j.jmarsys.2009.12.015.

Banas, N. S. (2011). Adding complex trophic interactions to a size-spectral plankton model: Emergent diversity paHerns and limits on predictability. Ecol. Modell. 222, 2663–2675. doi:10.1016/j.ecolmodel.2011.05.018.

Bonachela, J. A., Klausmeier, C. A., Edwards, K. F., Litchman, E., and Levin, S. A. (2016). The role of phytoplankton diversity in the emergent oceanic stoichiometry. J. Plankton Res. 38, 1021–1035. doi:10.1093/plankt/qv087.

Bruggeman, J., and Kooijman, S. A. L. M. (2007). A biodiversity-inspired approach to aquatic ecosystem modeling. Limnol. Oceanogr. 52, 1533–1544. doi:10.4319/lo.2007.52.4.1533.

Chen, B., Smith, S. L., and Wirtz, K. W. (2019). Effect of phytoplankton size diversity on primary productivity in the North Pacific: trait distributions under environmental variability. Ecol. Le=. 22, 56–66. doi:10.1111/ele.13167.

Clark, J. R., Daines, S. J., Lenton, T. M., Watson, A. J., and Williams, H. T. P. (2011). Individualbased modelling of adaptation in marine microbial populations using genetically defined physiological parameters. Ecol. Modell. 222, 3823–3837. doi:10.1016/j.ecolmodel.2011.10.001.

Daines, S. J., Clark, J. R., and Lenton, T. M. (2014). Multiple environmental controls on phytoplankton growth strategies determine adaptive responses of the N:P ratio. Ecol. Le=. 17, 414–425. doi:10.1111/ele.12239.

Dutkiewicz, S., Cermeno, P., Jahn, O., Follows, M. J., Hickman, A. E., Taniguchi, D. A. A., et al. (2020). Dimensions of marine phytoplankton diversity. Biogeosciences 17, 609–634. doi:10.5194/bg-17-609-2020.

Follows, M. J., and Dutkiewicz, S. (2011). Modeling diverse communities of marine microbes. Ann. Rev. Mar. Sci. 3, 427–451. doi:10.1146/annurev-marine-120709-142848.

Follows, M. J., Dutkiewicz, S., Grant, S., and Chisholm, S. W. (2007). Emergent biogeography of microbial communities in a model ocean. Science. 315, 1843–1846. doi:10.1126/science.1138544.

Hellweger, F. L., and Bucci, V. (2009). A bunch of tiny individuals-Individual-based modeling for microbes. Ecol. Modell. 220, 8–22. doi:10.1016/j.ecolmodel.2008.09.004.

Hellweger, F. L., Clegg, R. J., Clark, J. R., Plugge, C. M., and Kreu, J. U. (2016). Advancing microbial sciences by individual-based modelling. Nat. Rev. Microbiol. 14, 461–471. doi:10.1038/nrmicro.2016.62.

Hellweger, F. L., and Kianirad, E. (2007). Individual-based modeling of phytoplankton: Evaluating approaches for applying the cell quota model. J. Theor. Biol. 249, 554–565. doi:10.1016/j.jtbi.2007.08.020.

Hochfeld, I., and Hinners, J. (2024). Evolutionary adaptation to steady or changing environments affects competive outcomes in marine phytoplankton. Limnol. Oceanogr. 69, 1172–1186. doi:10.1002/lno.12559.

Kerimoglu, O., Hofmeister, R., Maerz, J., Riethmüller, R., and Wirtz, K. W. (2017). The acclimative biogeochemical model of the southern North Sea. Biogeosciences 14, 4499–4531. doi:10.5194/bg-14-4499-2017.

Kiørboe, T., and Andersen, K. H. (2019). Nutrient affinity, half-saturation constants and the cost of toxin production in dinoflagellates. Ecol. Le=. 22, 558–560. doi:10.1111/ele.13208.

Klausmeier, C. A., Kremer, C. T., and Koffel, T. (2020). 'Trait-based ecological and ecoevolutionary theory', in TheoreBcal Ecology: concepts and applicaBons, eds. K. S. McCann and G. Gellner (Oxford University Press).

Kremer, C. T., Thomas, M. K., and Litchman, E. (2017a). Temperature- and size-scaling of phytoplankton population growth rates: reconciling the Eppley curve and the metabolic theory of ecology. Limnol. Oceanogr. doi:10.1002/lno.10523.

Kremer, C. T., Williams, A. K., Finiguerra, M., Fong, A. A., Kellerman, A., Paver, S. F., et al. (2017b). Realizing the potential of trait-based aquatic ecology: New tools and collaborative approaches. Limnol. Oceanogr. 62, 253–271. doi:10.1002/lno.10392.

Litchman, E., and Klausmeier, C. A. (2008). Trait-based community ecology of phytoplankton. Annu. Rev. Ecol. Evol. Syst. 39, 615–639. doi:10.1146/annurev.ecolsys.39.110707.173549.

Merico, A., Bruggeman, J., and Wirtz, K. (2009). A trait-based approach for downscaling complexity in plankton ecosystem models. Ecol. Modell. 220, 3001–3010. doi:10.1016/j.ecolmodel.2009.05.005.

Norberg, J. (2004). Biodiversity and ecosystem functioning: A complex adaptive systems approach. Limnol. Oceanogr. 49, 1269–1277. doi:10.4319/lo.2004.49.4_part_2.1269.

Norberg, J., Urban, M. C., Vellend, M., Klausmeier, C. a., and Loeuille, N. (2012). Ecoevolutionary responses of biodiversity to climate change. Nat. Clim. Chang. 2, 747–751. doi:10.1038/nclimate1588.

Pahlow, M., Vézina, A. F., Casault, B., Maass, H., Malloch, L., Wright, D. G., et al. (2008). Adaptive model of plankton dynamics for the North Atlantic. Prog. Oceanogr. 76, 151– 191. doi:10.1016/j.pocean.2007.11.001.

Smith, S. L., Pahlow, M., Merico, A., Acevedo-Trejos, E., Sasai, Y., Yoshikawa, C., et al. (2015). Flexible phytoplankton functional type (FlexPFT) model: size-scaling of traits and optimal growth. J. Plankton Res. 38, 977–992. doi:10.1093/plankt/qv038.

Smith, S. L., Pahlow, M., Merico, A., and Wirtz, K. W. (2011). Optimality-based modeling of planktonic organisms. Limnol. Oceanogr. 56, 2080–2094. doi:10.4319/lo.2011.56.6.2080.

Taherzadeh, N., Kerimoglu, O., and Wirtz, K. W. (2017). Can we predict phytoplankton community size structure using size scalings of eco-physiological traits? Ecol. Modell. 360, 279–289. doi:10.1016/j.ecolmodel.2017.07.008.

Terseleer, N., Bruggeman, J., Lancelot, C., and Gypens, N. (2014). Trait-based representation of diatom functional diversity in a plankton functional type model of the eutrophied Southern North Sea. Limnol. Oceanogr. 59, 1–16. doi:10.4319/lo.2014.59.6.0000.

Thomas, M. K., Kremer, C. T., Klausmeier, C. A., and Litchman, E. (2012). A global pattern of thermal adaptation in marine phytoplankton. Science. 338, 1085–8. doi:10.1126/science.1224836.

Toseland, A., Daines, S. J., Clark, J. R., Kirkham, A., Strauss, J., Uhlig, C., et al. (2013). The impact of temperature on marine phytoplankton resource allocation and metabolism. Nat. Clim. Chang. 3, 1–6. doi:10.1038/nclimate1989.

Ward, B. A., Collins, S., Dutkiewicz, S., Gibbs, S., Bown, P., Ridgwell, A., et al. (2019). Considering the Role of Adaptive Evolution in Models of the Ocean and Climate System. J. Adv. Model. Earth Syst. 11, 3343–3361. doi:10.1029/2018MS001452.

Ward, B. A., Dutkiewicz, S., Jahn, O., and Follows, M. J. (2012). A size-structured food-web model for the global ocean. Limnol. Oceanogr. 57, 1877–1891. doi:10.4319/lo.2012.57.6.1877.

Wirtz, K. W. (2013). Mechanistic origins of variability in phytoplankton dynamics: Part I: niche formation revealed by a size-based model. Mar. Biol. 160, 2319–2335. doi:10.1007/s00227-012-2163-7.

Zakharova, L., Meyer, K. M., and Seifan, M. (2019). Trait-based modelling in ecology: A review of two decades of research. Ecol. Modell. 407, 108703. doi:10.1016/j.ecolmodel.2019.05.008.

Associate editor:

Dear authors,
Thank you for your responses to the reviewers' comments. As you have seen both referees have found it difficult to understand the difference between the model used in this paper and the one used in your previous study published in L&O. Hence both reviewers indicate the need to clarify what has been modified, or expanded, in the new article relative to the previous one. They also find that there is significant overlap and duplication between the two papers and that the conclusions are similar, although the focus in the present submission are the ecosystem functions.
Reviewer 1 stresses the need to improve the narrative structure of the manuscript, focusing both in Results and Discussion in results of broad significance rather than analysing in detail the quantitative response of each model species. This reviewer also emphasizes the need to note that projected ecosystem changes could be different if the model were to represent also changes in stratification/mixing and not just temperature.
Reviewer 2 notes that it is difficult to ascertain the novelty of the modelling approach as currently described and finds that the manuscript lacks a thorough presentation of the assumptions and limitations of the model. This reviewer also indicates that the Introduction should provide a broader context for the study by considering previous efforts to model the adaptive responses of phytoplankton to environmental changes.
In view of these comments, I invite you to prepare a thoroughly revised version of your manuscript, which should address all the issues indicated above. Both reviewers have expressed their willingness to examine a revised version of the article. Therefore, the revised manuscript will be sent to both reviewers for additional revision.
The new manuscript should clarify what is new in this contribution, both in terms of model configuration and analysis and, critically, in terms of new knowledge and insight obtained.
It is important that the article stands on its own – the model configuration and assumptions need not be described in their entirety (as this has been done already in the previous article) but the limitations and biases of the model as far as ecosystem functions are concerned should

be addressed more fully in the Discussion.

The Introduction should acknowledge thoroughly previous modelling studies that have considered evolution of plankton traits, and state more clearly what is new about the modelling approach used here. Biogeosciences does not have a limit of three references to support one statement.

Thank you for submitting your work to Biogeosciences.

Best regards,

Emilio Marañón

**2. Author's response**

Answer to Reviewer 1:

We would like to thank the reviewer very much for this helpful and constructive feedback and would be happy to revise our manuscript to take their understandable points of criticism into account. As the reviewer correctly pointed out, this study is largely based on a model we have already published in L&O. However, while the previously published study focused on the analysis of phytoplankton population dynamics, in this manuscript we focus on the larger, ecological aspects. To introduce readers to the model, we chose to summarize the results from our published study (lines 230−254), as these results are also relevant to the discussion of the new ecological results. However, we understand the criticism that this makes it unclear what has already been published and what is new in this manuscript. We therefore suggest moving Fig. 2, which can also be found in a similar form in Hochfeld & Hinners (2024), to the appendix and formulating a clearer delineation of the new results compared to those previously published. Similarly, in the revised version of our manuscript, we suggest describing the modifications of the model more clearly compared to the previously published model. Finally, we propose to fundamentally revise the manuscript in order to emphasize the new key general findings more clearly.

References

Hochfeld, I., & Hinners, J. (2024). Evolutionary adaptation to steady or changing environments affects competitive outcomes in marine phytoplankton. Limnology and Oceanography, 69(5), 1172–1186. https://doi.org/10.1002/lno.12559

Answer to Reviewer 2:

We thank the reviewer for the extensive feedback on our manuscript. Regarding the reviewer's first criticism: The model we presented is largely identical to the model already published in Hochfeld & Hinners (2024). This means that the simulated phytoplankton population dynamics are identical in the previously published version and in the version presented in this manuscript. However, to be able to analyze ecosystem functions, we had to modify the output parameters to be able to estimate carbon export, nitrogen fixation, and resource use efficiency (RUE). Regarding RUE, we moreover had to exclude the zooplankton and the cyanobacteria

from the simulations, as their grazing and nitrogen input, respectively, would make a meaningful calculation impossible. Currently, this is described in lines 179−198. We propose not to generally refer to the model as "modified" or "extended" in a future version of this manuscript to avoid confusion, but to clearly describe the model modifications in comparison to the published model.

Regarding the second criticism of the reviewer: As described in the response to reviewer 1, we understand that our motivation to introduce the model to readers by summarizing the results of the published article (Hochfeld & Hinners, 2024) has led to confusion about what is new in this manuscript. We therefore suggest moving Fig. 2 (which, as reviewer 2 points out, overlaps with the results of Hochfeld & Hinners, 2024) to the appendix and focusing our manuscript more clearly on the new results on ecosystem functioning. Since our model, as described above, is largely identical to the previously published model, we consider it more appropriate to focus the model description in this manuscript on the modifications rather than, as the reviewer suggests, to describe the model again in detail, which we have already done extensively in the previously published version (12 pages in the supplementary material). Unfortunately, we cannot follow the reviewer's criticism of our lack of discussion of model limitations. We discuss the assumptions and limitations of our model very specifically for the different aspects examined in the discussion (lines 411−412, 466−469, 480−482, 502−513, 542−545, and 581−584). We can only assume that the reviewer did not find this clear enough. We suggest that in a revised version of the manuscript, we summarize the model biases more clearly in a separate paragraph in the discussion. If we are invited to submit a revised version of this manuscript, we would be thankful for a clear guidance by the editor how to deal with these criticisms regarding model description and limitations, whether to follow our suggested modifications or the modifications asked for by the reviewer.

Regarding the reviewer's third criticism: We thank the reviewer for the extensive list of articles that investigated phytoplankton diversity dynamics. As far as we understand, most of these models deal with diverse phytoplankton populations and thus allow for one aspect of evolution: selection from a diverse pool of individuals. The second important aspect of evolution, the possibility of new mutations, is not taken into account by almost all of the models mentioned. In our introduction, we focus on the ecosystem modeling studies that take both aspects of evolution into account. This is still a new approach that has only been implemented in a few studies so far (e.g., Beckmann et al., 2019; Hinners et al., 2019; Le Gland et al., 2021; Sauterey et al., 2017; Smith et al., 2016). In the revised version, we propose to include a more detailed description of models that investigate phytoplankton diversity dynamics mentioned by the reviewer. In previous review processes, we were advised to support each statement with a maximum of three references. We found no clear information on this for Biogeosciences. We would be grateful for a brief statement from the editor on the maximum number of references that should be given per statement.

Regarding the final criticism of the reviewer: The available long-term data for the Baltic Sea are unfortunately not sufficient for an extensive quantitative model calibration and validation. Data on species level are sparse and provide insufficient temporal resolution/coverage to calibrate or validate our model with respect to bloom timing and relative abundances of our

focal species. Instead, we used data on functional group level from Hjerne et al. (2019) to validate the model qualitatively regarding bloom timing and relative abundances of phytoplankton groups. We suggest including a subsection on model validation into the results section and a discussion of the missing quantitative calibration and validation into our description of model limitations.

References

Beckmann, A., Schaum, C.-E., & Hense, I. (2019). Phytoplankton adaptation in ecosystem models. Journal of Theoretical Biology, 468, 60–71. https://doi.org/10.1016/j.jtbi.2019.01.041

Hinners, J., Hense, I., & Kremp, A. (2019). Modelling phytoplankton adaptation to global warming based on resurrection experiments. Ecological Modelling, 400, 27–33. https://doi.org/10.1016/j.ecolmodel.2019.03.006

Hjerne, O., Hajdu, S., Larsson, U., Downing, A. S., & Winder, M. (2019). Climate Driven Changes in Timing, Composition and Magnitude of the Baltic Sea Phytoplankton Spring Bloom. Frontiers in Marine Science, 6. https://www.frontiersin.org/articles/10.3389/fmars.2019.00482

Hochfeld, I., & Hinners, J. (2024). Evolutionary adaptation to steady or changing environments affects competitive outcomes in marine phytoplankton. Limnology and Oceanography, 69(5), 1172–1186. https://doi.org/10.1002/lno.12559

Le Gland, G., Vallina, S. M., Smith, S. L., & Cermeño, P. (2021). SPEAD 1.0 – Simulating Plankton Evolution with Adaptive Dynamics in a two-trait continuous fitness landscape applied to the Sargasso Sea. Geoscientific Model Development, 14(4), 1949–1985. https://doi.org/10.5194/gmd-14-1949-2021

Sauterey, B., Ward, B., Rault, J., Bowler, C., & Claessen, D. (2017). The Implications of EcoEvolutionary Processes for the Emergence of Marine Plankton Community Biogeography. The American Naturalist, 190(1), 116–130. https://doi.org/10.1086/692067

Smith, S. L., Vallina, S. M., & Merico, A. (2016). Phytoplankton size-diversity mediates an emergent trade-off in ecosystem functioning for rare versus frequent disturbances. Scientific Reports, 6(1), 34170. https://doi.org/10.1038/srep34170

**3. Author's changes in the manuscript**

Dear editor and dear reviewers,

We tried to incorporate all your comments and suggestions; below, we list the changes we made to our manuscript. All line numbers refer to the marked-up version of our manuscript. In addition to addressing all your comments, we tried to improve the overall clarity of our manuscript and to focus more on the broad qualitative results.

We would like to thank you for your helpful comments and suggestions, as well as all persons involved in processing this manuscript. We honestly hope that you agree with our revision and look forward to hearing from you.

Kind regards

Isabell Hochfeld and Jana Hinners

**List of changes in the manuscript**

1 Introduction

We tried to focus the introduction more on ecosystem functioning instead of phytoplankton. For this purpose, we rewrote l. 45-65.

l. 103-120: We added some of the studies suggested by Reviewer 2 to provide a broader context for our study.

In addition, we tried to increase the focus of our introduction and thus made additional small changes.

2 Materials & methods

We tried to make a clearer distinction between the original model from Hochfeld and Hinners (2024) and the modifications we made to the model in this study. In the revised version of our manuscript, we describe the original model in Sect. 2.1 and our modifications in Sect. 2.2.

We rewrote Sect. 2.1 as a summary of the original model used in Hochfeld and Hinners (2024). To do so, we not only made changes to the text but also moved Fig. 1 to the Appendix.

In Sect. 2.2, we describe the modifications that we made to the original model, i.e., the explicit calculation of different ecosystem functions (carbon export, nitrogen fixation, and resource use efficiency). To make this clearer, we renamed the section from "Ecosystem functions" to "Model modifications" and rewrote the first paragraph (l. 206-208). Since our modifications also include the change of a single model parameter, we added the details to l. 216-218.

In Sect. 2.3, we tried to make clearer that we adopted our model scenarios from Hochfeld and Hinners (2024). In addition, we deleted a paragraph that overlaps significantly with our previous paper (l.250-256) and removed the corresponding figure from the Appendix (Fig. B1).

3 Results

We rewrote this section to reduce overlap with Hochfeld and Hinners (2024) and to focus on the new key results.

We merged Sects. 3.1 and 3.2 into a new section ("Model validation") in which we qualitatively validate our modeled seasonal phytoplankton and zooplankton dynamics against observations. For this purpose, we deleted our summary of the results from Hochfeld and

Hinners (2024) (l. 274-287) and our detailed comparison of zooplankton dynamics between the four model scenarios (l. 303-327). In addition, we moved a modified version of Fig. 2, Fig. 3, and a modified version of Table 2 to the Appendix (now Table A2, Figs. B2 and B3). In the modified version of Fig. 2, we visualized the observed bloom periods in the Baltic Sea (spring, summer, and autumn), which we derived from Hjerne et al. (2019). In the modified version of Table 2, we replaced the information on zooplankton peak abundance, which appears irrelevant in the revised version of the manuscript, by information on phytoplankton timing in spring.

Section 3.2 (previously 3.3), which describes the annual balances of the different ecosystem functions, is the new focus of our revised manuscript. To focus more on the broad qualitative results, we rewrote especially the paragraph on primary production (l. 356-371) and removed Table 3. In addition, we added a figure to the Appendix that shows more clearly the contrasting development of dinoflagellates compared to cyanobacteria and diatoms (Fig. B4).

4 Discussion

In Sect. 4.1, we tried to focus more on primary production as a whole instead of the detailed phytoplankton community dynamics. For this purpose, we renamed Sect. 4.1 from "Phytoplankton" to "Primary production" and deleted the detailed discussion on dinoflagellate and diatom dynamics (l. 464-508). In addition, we tried to focus more on the effects of adaptation on primary production (l. 449-458). For this purpose, we included a summary of the effects of adaptation on phytoplankton dynamics found by Hochfeld and Hinners (2024) (l. 450-456) to explain our results and to prepare the reader for the discussion of the remaining ecosystem functions.

We moved our discussion of nitrogen fixation from Sect. 4.4 to Sect. 4.2 since we think it fits better after our discussion of primary production. Furthermore, we extended our discussion of the effects of adaptation on nitrogen fixation since adaptation is the focus of our study (l. 526-529).

We renamed Sect. 4.3 (former 4.2) from "Zooplankton" to "Secondary production". In addition, we again tried to emphasize the role of adaptation (l. 545-549) and notably shortened the section to focus on the broad results. For this purpose, we deleted our detailed discussion of the time lag between phytoplankton and zooplankton (l. 558-562) and moved everything related to model biases and suggestions for future work (l. 563-571, 578-580, and 584-587) to the newly created Sect. 4.7 ("Model biases").

Regarding Sect. 4.4 ("Carbon export", former 4.3), we moved the paragraph on model biases (l. 605-618) to Sect. 4.7. We moreover added a short paragraph to put our results into the broader context (l. 600-604).

We strongly shortened and rewrote Sect. 4.5 ("Resource use efficiency (RUE)") to focus our discussion more on the broad qualitative results instead of the detailed quantitative results. In addition, we moved the part on biases and limitations of our RUE calculations (l. 693-697) to Sect. 4.7.

We summarized Sect. 4.6 ("Control factors and feedbacks in our model ecosystem") to better highlight the key dynamics and mechanisms in our model ecosystem without getting lost in details.

We added Sect. 4.7 ("Model biases and outlook") to our manuscript to discuss the assumptions, limitations, and biases of our model regarding ecosystem functioning in more detail than before. In this section, we discuss the following aspects:

1. The lack of a quantitative model validation
2. Biases in RUE due to species-specific constraints on adaptation
3. Biases in zooplankton due to simplistic representation
4. Biases due to 0-dimensional model setup

5 Conclusions

We tried to better highlight our main conclusions.

Appendices

We removed Table A1 from Appendix A and added a modified version of Table 2 (now Table A2, we replaced zooplankton peak abundance by phytoplankton timing in spring).

We removed Figs. B1, B2, B4, and B5 from Appendix B and added Figs. 1 (now B1), 2 (now B2, we included bloom periods from Hjerne et al. 2019), 3 (now B3), and a new Fig. B4 (shows the annual balances of dinoflagellates, diatoms, and cyanobacteria in separate panels to make the contrasting development of dinoflagellates clearly visible).

References

Hjerne, O., Hajdu, S., Larsson, U., Downing, A. S., & Winder, M. (2019). Climate Driven Changes in Timing, Composition and Magnitude of the Baltic Sea Phytoplankton Spring Bloom. *Frontiers in Marine Science*, *6*. https://www.frontiersin.org/articles/10.3389/fmars.2019.00482

Hochfeld, I., & Hinners, J. (2024). Evolutionary adaptation to steady or changing environments affects competitive outcomes in marine phytoplankton. *Limnology and Oceanography*, *69*(5), 1172–1186. https://doi.org/10.1002/lno.12559

---

## Author Response (AR2)

**Author's response**

**1. Comments from the referees**

Anonymous reviewer:

I thank the authors for the amendments implemented in their manuscript, which help better understand their findings and distinguish them from their previous work. I have one aspect that I would appreciate if the authors could address in the discussion of model limitations. How does their implementation of adaptive/flexible response (temperature-related traits) bias or impact their interpretation/prediction and how does it differ from others (e.g. Norberg et al. 2012 and others just cited in the introduction but not discussed)?

Norberg, J., Urban, M. C., Vellend, M., Klausmeier, C. a., & Loeuille, N. (2012). Eco-evolutionary responses of biodiversity to climate change. Nature Climate Change, 2(10), 747–751. https://doi.org/10.1038/nclimate1588

Associate editor:

Dear authors,

Your revised manuscript has now been examined by one of the original reviewers, who finds that the new version makes it easier to understand your findings and clarifies the differences with your earlier article. The revised manuscript also includes a new section on biases and limitations of your model, which can help future efforts to investigate the implications of phytoplankton adaptation to global change. I am therefore recommending publication of your article subject to a minor revision, namely the inclusion in the Discussion of some comments regarding how your approach to model adaptative responses differs from those of other studies cited (e.g. Norberg et al. 2012).

Thank you for submitting your work to Biogeosciences.

Best regards,

Emilio Marañón

**2. Author's response/changes in the manuscript**

Dear editor and dear reviewer,

As suggested, we have added a paragraph to our discussion of model biases (Sect. 4.7) in which we discuss possible biases of our evolutionary algorithm and point out the differences between our approach and other studies cited in the Introduction (l. 508-536). To better connect this paragraph to the Introduction, we also made changes to the paragraph about evolution and ecosystem modeling in the Introduction (l.80-105). In addition, we added a copyright statement to the caption of Fig. B1 and indicated that the resuspension process

shown in the figure was disabled in our simulations (l. 588-598). Line numbers refer to the marked-up version of our manuscript.

We would like to thank you again for your constructive and helpful feedback, as well as all the people involved in processing this manuscript. We sincerely hope that you agree with our updated revision and look forward to hearing from you.

Kind regards

Isabell Hochfeld and Jana Hinners